# Time Series Forecasting as Reasoning: A Slow-Thinking Approach with Reinforced LLMs

## Abstract

To advance time series forecasting (TSF), various methods have been proposed to improve prediction accuracy, ranging from statistical techniques to data-driven deep learning architectures. Despite their effectiveness, most existing methods still adhere to a *fast thinking* paradigm-relying on extracting historical patterns and mapping them to future values as their core modeling philosophy, lacking an explicit reasoning process. Meanwhile, emerging slow-thinking LLMs (e.g., OpenAI-o1) have shown remarkable multi-step reasoning capabilities, offering an alternative way to overcome these issues. However, prompt engineering alone presents several limitations—including high computational cost, privacy risks, and limited capacity for in-depth domain-specific time series reasoning. To address these limitations, a more promising approach is to train LLMs to develop *slow thinking* capabilities and acquire strong time series reasoning skills. For this purpose, we propose Time-R1, a two-stage reinforcement fine-tuning framework designed to enhance multi-step reasoning ability of LLMs for time series forecasting. Specifically, the first stage conducts supervised fine-tuning for warmup adaptation, while the second stage employs reinforcement learning to improve the model's generalization ability. Particularly, we design a fine-grained multi-objective reward specifically for time series forecasting, and then introduce GRIP (group-based relative importance for policy optimization), which leverages non-uniform sampling to further encourage and optimize the model's exploration of effective reasoning paths. Experiments shows Time-R1 significantly improves forecast performance in diverse datasets [1].

## 1 Introduction

Time series forecasting (TSF) (Cheng et al., 2025b;c) plays a key role in data-driven decision-making across critical domains, including financial market analysis, energy demand planning, and traffic flow management. Over the years, numerous research efforts have been proposed to advance this field. Classical statistical methods such as ARIMA (Zhang, 2003), ETS (Hyndman et al., 2008), and Theta (Assimakopoulos & Nikolopoulos, 2000) have long been used to predict future data by leveraging the statistical properties of single samples. Machine learning methods such as XGBoost(Chen & Guestrin, 2016) and LightGBM(Ke et al., 2017) remain highly robust due to their interpretability and ability to model nonlinear relationships. With the arrival of computing power era, deep learning-based approaches (Cheng et al., 2025a; Wang et al., 2024a) have since gained prominence due to their ability to capture complex temporal patterns and adapt to non-stationary real-world data. Methodological analysis covers pioneering architectures such as sequence dependency modeling of RNNs (Hewamalage et al., 2021; Salinas et al., 2020), TCNs (Hewage et al., 2020; Cheng et al., 2025e) and transformer-based models (Cheng et al., 2023), which improve generalization through shared representations across multiple time series.

Although the specific techniques vary, most existing TSF methods follow a similar "fast thinking" paradigm (Wang et al., 2024a; 2025; Liu et al., 2025a). Specifically focusing on single-step prediction accuracy (Liu et al., 2024c), these methods typically employ sequential models to encode historical values and use one-step decoding to directly map past observations to future values (Cheng et al., 2025e). Although effective in benchmarks, their underlying logic is largely based on pattern recognition (Cheng et al., 2025a) and trend prediction, lacking an explicit reasoning process. However, in

---

[1]The code is at `https://anonymous.4open.science/r/Time-R1-ICLR-2026`.

Figure 1: Overview of evolution of TSF methods. Time-R1 is a novel, general forecasting paradigm.

real-world scenarios, time series often reflect more complex temporal logic, which should not merely be 'fitted'—they should be understood and reasoned.

To address this issue, a growing body of research (Chang et al., 2023; 2024) has explored leveraging the reasoning capabilities of large language models (LLMs) to analyze temporal dynamics and generate high-quality representations, thereby enhancing lightweight TSF models (Liu et al., 2025b; Jin et al., 2023b). These approaches benefit from the ability of LLMs to incorporate contextual information such as textual metadata (Gruver et al., 2023), offering stronger generalization performance across diverse domains (Liu et al., 2024b; Dooley et al., 2023), and produce explanations that support forecasting decisions (Tan et al., 2024; Cheng et al., 2025a). However, despite their potential, current LLM-based TSF methods face three key limitations: *First*, a partial misalignment of time series domain knowledge, and limited reasoning capabilities. General linguistic knowledge in LLM often mismatches the temporal patterns and causal mechanisms required for time series tasks, leading to suboptimal performance (Zhou et al., 2023; Jin et al., 2023a). *Second*, a lack of generalization from experiential learning. While effective in supervised memorization, they struggle with understanding dynamics or adapting to new, unseen scenarios, which limits their out-of-distribution performance. *Third*, absence of progressive reasoning. These models map history to future directly without detecting regime changes or performing step-by-step inference, resembling fast (not deliberate) thinking for time series. These issues lead to a central question: **Can we improve time series forecasting performance by training LLMs to acquire time series reasoning capabilities?**

Motivated by the above question, we propose Time-R1, a novel LLM-based time series forecasting framework that trains large language models to acquire slow-thinking reasoning capabilities for forecasting tasks. At its core, Time-R1 leverages LLMs as the time series reasoning backbone and introduces a two-stage reinforcement fine-tuning (RFT) optimization framework: *First*, we begin with warm-up supervised fine-tuning. The model is fine-tuned for memorization, learning both effective reasoning patterns and accurate output formatting using synthetic reasoning trajectories that demonstrate step-by-step temporal analysis. *Second*, the model is refined through reinforcement learning for generalization, using fine-grained, multi-objective rewards specifically designed for forecasting tasks, improving temporal coherence and multi-horizon accuracy. Notably, we propose GRIP (Group-based Relative Importance for Policy Optimization), which optimizes LLM reasoning paths in TSF through a uniform sampling strategy and adaptive weighting. Extensive experiments are conducted on real-world datasets, showing that Time-R1 effectively enhances forecast performance through the slow thinking paradigm. Our main contributions are as follows:

- We introduce time series reasoning by training LLMs to adopt a slow-thinking paradigm that generates reasoning processes supporting final forecasting.
- We design a two-stage RFT framework (SFT for memorization, and RL for generalization) that enhances the reasoning ability of LLMs. We introduce a fine-grained, multi-objective reward specifically for TSF, along with a novel sampling strategy for RL optimization.
- Extensive experiments demonstrate the effectiveness of Time-R1, showing it enhances LLM reasoning and improves generalization and explainability via deliberate slow thinking.

## 2 PRELIMINARIES

### 2.1 PROBLEM DEFINITION

Let $\mathbb{D} = \{(X^i, y^i)\}_{i=1}^n$ be a temporal dataset, where each $X^i \in \mathbb{R}^{t \times m}$ is a multivariate time series with $t$ steps and $m$ channels, and $y^i \in \mathbb{R}^{h \times d}$ contains $d$-dimensional targets over $h$ future steps. The forecasting task learns a mapping $f_\theta : \mathbb{R}^{t \times m} \to \mathbb{R}^{h \times d}$ capturing temporal dependencies in $\mathbb{D}$. Under Time-R1, the forecasting procedure using prompt template $P$ is: $T^i = \text{LLM}_\phi(P, X^i)$, where $T^i$ is the LLM's textual output, and $\hat{y}^i = g(T^i)$ parses it into numerical predictions $\hat{y}^i \in \mathbb{R}^{h \times d}$.

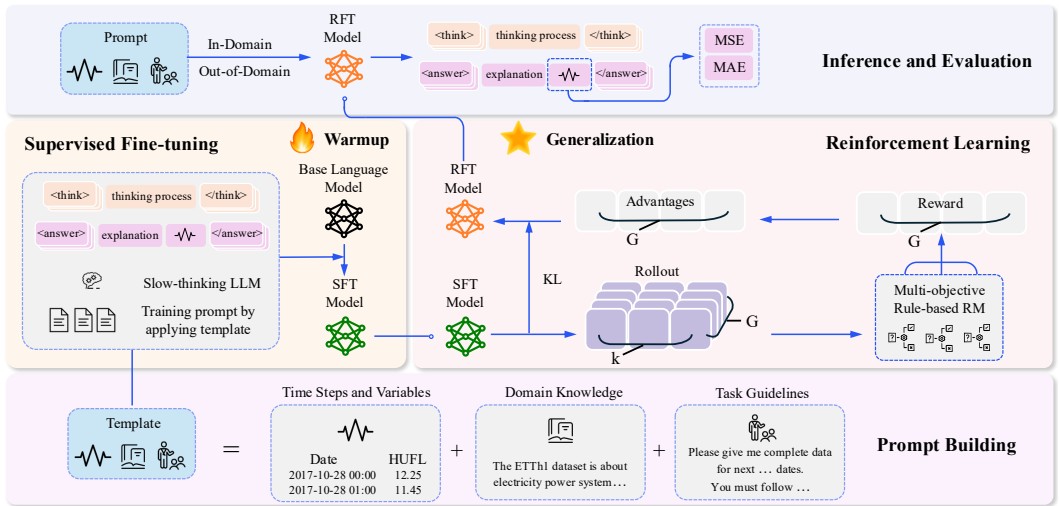

Figure 2: A diagram illustrating the three steps of Time-R1: (1) building a training template with domain context, time steps, and variables, (2) collecting long-CoT data from DeepSeek-R1 using the template to train a supervised policy, and (3) optimizing the policy via reinforcement learning with group-based relative importance for policy optimization (GRIP) to enhance TSF reasoning capability.

## 2.2 TIME SERIES REASONING

Despite rapid advances in time series forecasting (TSF), most methods lack genuine *time series reasoning*—the ability to dynamically infer future values by integrating historical data with structured logic or domain knowledge. Traditional statistical (Hyndman et al., 2008; Zhang, 2003) and classical machine learning models (Ke et al., 2017; Chen & Guestrin, 2016) are limited in capturing high-level temporal semantics, while deep learning approaches (Cheng et al., 2023; Hewage et al., 2020; Hewamalage et al., 2021) often act as "black boxes" without interpretable reasoning. Although LLM-based methods (Chang et al., 2024; 2023; Kong et al., 2025) have introduced language-like interpretability, they typically rely on prompt engineering and general-purpose models not designed for temporal reasoning, constraining their capacity for deliberate inference. Recent reasoning-oriented LLMs (e.g., OpenAI-o1/o3 (OpenAI, 2025), DeepSeek-R1 (Guo et al., 2025)) offer promise through "slow thinking" mechanisms, but remain computationally heavy, ill-suited for temporal tasks, and impractical for privacy-sensitive domains. Thus, prompt engineering alone is insufficient, and there is a critical need for a dedicated framework that explicitly trains LLMs on structured forecasting tasks to develop domain-specific, robust time series reasoning capabilities.

## 3 THE PROPOSED TIME-R1

### 3.1 TIME-R1 OVERVIEW

Time-R1 consists of a two-stage RFT framework for LLM-based time series forecasting, built upon a structured training template that standardizes input representations and encodes task-specific knowledge. In the first stage, we perform warm-up SFT using synthetic chain-of-thought trajectories to teach the model effective temporal analysis and accurate output formatting. These trajectories are generated under strict guidelines and refined iteratively to align with ground-truth forecasts. The second stage further improves the model via RL, guided by a fine-grained, multi-objective reward function tailored for time series forecasting. To optimize reasoning paths during RL, we introduce GRIP (Group-based Relative Importance for Policy Optimization), a novel strategy that leverages non-uniform sampling and adaptive weighting to balance accuracy, logical consistency, and temporal coherence. An overview of the framework is provided in Figure 2.

### 3.2 TRAINING TEMPLATE

Our training template employs an instructional framework that standardizes inputs while encoding task-specific knowledge through five components: (1) *Task Definition* establishing objectives and

problem scope; (2) *Dataset Description* specifying temporal characteristics and application scenarios; (3) *Channel Information* delineating input signal types; (4) *Testing Data* providing timestamps and historical series; and (5) *Format Instruction* defining output templates. This design interleaves domain knowledge with structural constraints, reducing inference-time formatting ambiguities (see Table 1).

Table 1: Training template for Time-R1. Contents enclosed in {} will be replaced with specific information, including dataset description, channel information, and historical time-series data.

> Here is the {Channel Information} data of the ETTh1 dataset. The ETTh1 dataset is {Dataset Description}. I will now give you data for the past 96 recorded dates, and please help me forecast the data for the next 96 recorded dates. The data is as follows: {Historical Time Series Data}. Please give me the complete data for the next 96 recorded dates. Remember to give me the complete data. You must first conduct reasoning inside <thinking>...</thinking>. When you have the final answer, you can output the answer inside <answer>...</answer>.

### 3.3 SUPERVISED FINE-TUNING FOR WARMUP ADAPTATION

To mitigate the linguistic readability degradation and slow convergence caused by direct reinforcement learning on LLMs—particularly due to formatting inconsistencies—we first perform a warmup stage via SFT. This warmup SFT step is designed to stabilize training, ensure proper output formatting, and equip the model with basic reasoning capabilities, without requiring deep time series understanding.

Our SFT data construction involves three key steps. First, we leverage DeepSeek-R1 to generate time-series predictions on the training set by feeding it historical time series data paired with strict formatting guidelines. We then select the optimal prediction for each sample based on the Mean Absolute Percentage Error metric De Myttenaere et al. (2016). Next, to derive a reasoning process aligned with ground-truth labels, we inject both the true prediction value and the high-quality CoT generated in the previous step into DeepSeek-R1 as prompts, guiding it to synthesize a revised CoT that logically culminates in the correct prediction. Finally, we concatenate the refined CoT and the true prediction value, demarcating the final answer using '<answer>' tags to create structured training data for SFT. The full data collection procedure is detailed in Algorithm 1 in the Appendix.

After constructing the training data, we perform a single-epoch fine-tuning with a small learning rate. This warm-up SFT phase effectively prepares the model for subsequent reinforcement learning, ensuring stable learning dynamics and accurate output formatting. It also enables the model to internalize reasoning patterns from synthetic trajectories, laying the foundation for more deliberate and coherent decision-making in later RL stages.

### 3.4 REINFORCEMENT LEARNING FOR EFFECTIVE REASONING PATTERNS

After warmup SFT, we further fine-tune the LLM using RL to generalize its reasoning and acquire slow-thinking capabilities for time series forecasting. In the following sections, we first present the reward design, and then describe the employed reinforcement learning algorithm GRIP.

#### 3.4.1 REWARD DESIGN

To effectively apply RL for optimizing the proposed slow-thinking time series forecasting, we introduce several fine-grained and multi-objective reward functions specifically designed to enhance forecasting performance and slow thinking behavior.

**Format Rewards.** To ensure syntactic validity and completeness of the generated reasoning paths, we define two reward components to enforce both structural integrity and output completeness: *Format Reward:* A binary penalty is imposed if the output does not follow the required structured format (e.g., missing or malformed `<answer>` tags):

$$\gamma_{\text{Format}} = \begin{cases} 0 & \text{if valid } \texttt{<think>}, \texttt{<answer>} \text{ tags and proper structure} \\ -1 & \text{otherwise} \end{cases} \tag{1}$$

*Length Reward:* To encourage full time point sequence generation and accelerate convergence, we provide positive feedback based on how close the generated sequence length is to the ground truth:

Figure 3: Overview of Group-based Relative Importance for Policy Optimization (GRIP).

$$\gamma_{\text{Length}} = \begin{cases} 0.1 & \text{if len(answer)} \geq \text{len(ground\_truth)} \\ 0.1 \cdot \dfrac{\text{len(answer)}}{\text{len(ground\_truth)}} & \text{otherwise} \end{cases} \tag{2}$$

**Accuracy Rewards.** We define two accuracy-based reward components to assess numerical precision and temporal fidelity, encouraging accurate value prediction and modeling of dynamics:

*MSE Reward:* We compute the mean squared error between normalized prediction and target sequences and map it into a bounded reward signal using a sigmoid transformation:

$$\gamma_{\text{MSE}} = \left(1 - \frac{1}{1 + e^{-0.3 \cdot \text{MSE}}}\right) \cdot 2, \tag{3}$$

*Seasonal-Trend Decomposition Reward:* Both predicted and true sequences are decomposed into seasonal ($s$) and trend ($t$) components via moving average-based methods, where $s_i$ the seasonal component and $t_i$ the long-trend component. We then compute separate MSE terms for each:

$$\gamma_{\text{Seasonal}} = \frac{1}{n} \sum_{i=1}^{n} \left(s_i^{\text{true}} - s_i^{\text{pred}}\right)^2, \gamma_{\text{Trend}} = \frac{1}{n} \sum_{i=1}^{n} \left(t_i^{\text{true}} - t_i^{\text{pred}}\right)^2, \tag{4}$$

**Structural Similarity Reward.** We evaluate structural similarity by matching predicted and ground-truth extrema within tolerance windows, ensuring change-point capture and interpretable patterns, with correct matches receiving credit:

$$\gamma_{\text{CP}} = \left(\frac{N_{\text{cmax}}}{N_{\text{gmax}}} \cdot 0.2\right) + \left(\frac{N_{\text{cmin}}}{N_{\text{gmin}}} \cdot 0.2\right), \tag{5}$$

where $N_{\text{cmax}}$ and $N_{\text{cmin}}$ respectively represent the counts of correctly identified local maxima and minima within a tolerance window, $N_{\text{gmax}}$ and $N_{\text{gmin}}$ are the total ground-truth extrema counts.

The overall reward for RL training is the sum of the above rewards.

### 3.4.2 REINFORCEMENT LEARNING ALGORITHM: GRIP

We introduce GRIP (Group-based Relative Importance for Policy Optimization) in Figure 3, a general RL optimization method designed to optimize entire trajectories for LLM time series forecasting reasoners. The GRIP objective function, formalized in Equation 6, combines a non-uniform sampling strategy with adaptive trajectory weighting within a policy gradient framework. In the following sections, we elaborate on its core components:(1) GRIP formalization; (2) non-uniform sampling strategy to balance exploration and exploitation; and (3) an adaptive weighting scheme that enhances gradient signals from high-quality reasoning paths.

**Formalization of the GRIP Objective.** The GRIP objective integrates the two key design components into a unified policy gradient framework, as formalized in Equation 6:

$$\mathcal{J}_{\text{GRIP}}(\theta) = \mathbb{E}_{\substack{q \sim P(Q), \\ \{o_j\}_{j=1}^{k \cdot G} \sim \pi_{\theta_{\text{old}}}(o|q), \\ \{o_i\}_{i=1}^{G} \sim \text{Sample}\left(\{o_j\}_{j=1}^{k \cdot G}; R(o_j)\right)}} \left\{ \sum_{i=1}^{G} w_i^U \frac{1}{|o_i|} \left\{ \sum_{t=1}^{|o_i|} \min\left[\frac{\pi_\theta(o_{i,t}|q, o_{i,<t})}{\pi_{\theta_{\text{old}}}(o_{i,t}|q, o_{i,<t})} A_i, \right.\right.\right.$$

$$\left.\left.\left. \text{clip}\left(\frac{\pi_\theta(o_{i,t}|q, o_{i,<t})}{\pi_{\theta_{\text{old}}}(o_{i,t}|q, o_{i,<t})}, 1 - \epsilon, 1 + \epsilon\right) A_i\right] - \beta \mathbb{D}_{KL}[\pi_\theta || \pi_{ref}]\right\}\right\}, \tag{6}$$

where, $\epsilon$ and $\beta$ are hyperparameters. $\pi_{\text{ref}}$ is the reference model, typically initialized as the pre-trained model before reinforcement learning begins. The output $\{o_i\}$ is selected through a sampling process from policy $\pi_{\theta_{\text{old}}}$. The hyperparameter $k$ controls the size of the rollout space, while $G$ referred to as the group size. $\mathbb{D}_{KL}$ represents the KL divergence, which is incorporated into the loss function as a regularization term during training. And $A_i$ is the advantage computed using a group of rewards $\{r_1, r_2, \ldots, r_G\}$ corresponding to the completion trajectories within each group:

$$A_i = \frac{r_i - \text{mean}(\{r_1, r_2, \ldots, r_G\})}{\text{std}(\{r_1, r_2, \ldots, r_G\})}, \tag{7}$$

The weight $w_i^U$ denotes the adaptive weighting assigned to each trajectory. This objective balances exploration and exploitation while mitigating gradient dilution. The sampling strategy and adaptive weight will be discussed in the following section.

**Non-uniform Sampling Strategy.**   To bridge the gap between reasoning and forecasting in time series modeling, recent RL methods like GRPO have shown promise. However, they often suffer from an exploration-exploitation imbalance. To address this, GRIP introduces a non-uniform sampling strategy that first generates $k \cdot G$ candidate trajectories $\{o_j\}_{j=1}^{k \cdot G}$ from policy $\pi_{\theta_{\text{old}}}$ (where $k$ scales exploration and $G$ is group size), then selects $G$ elite trajectories via reward-weighted sampling Sample($\{o_j\}; R(o_j)$). These are replicated to form the update set $\{o_i\}_{i=1}^{G}$, maintaining GRPO's update scale while emphasizing high-reward regions. Mathematically equivalent to importance-sampled policy gradient correction, this approach balances broad exploration with computational efficiency through its dual-phase design. To further generalize, GRIP supports two sampling strategies:

(1) *Local Random Sampling*: For each input question $q$, we first generate $k$ candidate trajectories $\{o_j\}_{j=1}^{k}$ by independently sampling from the old policy $\pi_{\theta_{\text{old}}}$. The trajectory with the highest reward $o^* = \arg\max_{1 \le j \le k} R(o_j)$ is selected as the elite sample. This process is repeated $G$ times to construct the final set $\{o_i\}_{i=1}^{G}$. This strategy emphasizes deterministic exploitation of the top-performing sample at each iteration while maintaining computational efficiency.

(2) *Cluster-based Random Sampling*: For each $q$, we generate $k \cdot G$ candidate trajectories $\{o_j\}_{j=1}^{k \cdot G}$. These trajectories are clustered based on their rewards (e.g., reward-binning or K-means clustering), and $G$ trajectories are randomly sampled across clusters to ensure diversity in the final update set. This method balances exploration and exploitation by preserving low-reward but potentially informative samples while still prioritizing high-reward paths.

**Adaptive Weighting for Gradient Enhancement.**   Traditional uniform weighting like GRPO ($1/G$) across trajectories fails to account for inter-trajectory quality disparities, leading to misleading gradients from low-quality samples and diminished signals for high-quality ones. GRIP addresses this by assigning trajectory-specific weights via softmax:

$$w_i^U = \frac{\exp(\hat{x}_{q,o_i})}{\sum_{j=1}^{G} \exp(\hat{x}_{q,o_j})}, \tag{8}$$

where the completion score $\hat{x}_{q,o_i}$ can be flexibly configured. For example, when $\hat{x}_{q,o_i} = R(o_i)$, the weighting amplifies the influence of high-reward trajectories. This adaptive weighting suppresses noise from low-quality outputs and strengthens gradients from critical trajectories.

## 4   EXPERIMENTAL SETUP

**Datasets and Evaluation Metrics.**   To ensure comprehensive evaluation across diverse scenarios, we conduct experiments in nine datasets spanning multiple domains with distinct temporal characteristics and data attributes (detailed in Table 4). These include: the ETT dataset (Zhou et al., 2021) capturing 2016-2018 electricity load records, Exchange (Lai et al., 2018) tracking 1990-2016 foreign exchange rates, Wind (Lai et al., 2018) with 2020-2021 wind measurements captured, AQ (Zhang et al., 2017) providing four-year air quality data, NASDAQ (Feng et al., 2019) provides complete stock market series including opening/closing prices, trading volumes, and daily high-low values. All datasets were evaluated using Mean Squared Error (MSE) and Mean Absolute Error (MAE) under a 96-step prediction setting, except NASDAQ which uses a 36-step configuration, with results reported as the MSE between predictions and ground truth. Data statistics is listed in Appendix E.

**Baselines.** Our baselines include competitive methods: PatchTST (Nie et al., 2022), DLinear (Zeng et al., 2023), FEDformer (Zhou et al., 2022), iTransformer (Liu et al., 2023), Autoformer (Wu et al., 2021), , TimeXer (Wang et al., 2024c), WPMixer (Murad et al., 2025) and TimeMixer (Wang et al., 2024b). We incorporate LLMs-based approaches, CrossTimeNet (Cheng et al., 2025d), GPT4TS (Zhou et al., 2023), TimeLLM (Jin et al., 2023a), and DeepSeek-R1 (Guo et al., 2025) for zeroshot.

**Implementation Details.** Unlike traditional TSF methods that often require normalization, we conduct experiments in the original numerical space. For Time-R1, we use Qwen2.5-7B-Instruct as the backbone. In SFT, we train on 300 synthetic samples with a learning rate of 5e-5 for one epoch. In RL, we implement GRIP using the Verl framework (Sheng et al., 2024) with vLLM for generation. In Eq. 6, $\epsilon = 0.2$ and $\beta = 0.04$; group size $G = 16$, $k = 3$. The batch size is 16, learning rate is 1e-6, policy temperature is 1, and max completion length is 3000. Both stages are run on a 4-GPU A800 cluster. For DeepSeek-R1, we apply its prompt directly to time series prediction without training. Time-R1 is trained only on ETTh1 and generalized to other datasets without fine-tuning, whereas baseline methods require separate models per dataset.

## 5 EXPERIMENTAL RESULTS

### 5.1 MAIN RESULTS

Table 2: Performance comparison of Time-R1 and baseline models with best values in bold and second-best underlined. MSE ↓ is used as the evaluation metric. Deepseek-R1 denotes zero-shot using our reasoning template. Overall, Time-R1 achieves superior performance across dataset.

| | Methods | ETTh1 | ETTh2 | ETTm1 | ETTm2 | Exchange | AQWan | AQShunyi | Wind | NASDAQ |
|---|---|---|---|---|---|---|---|---|---|---|
| Traditional | PatchTST | 9.3001 | 10.9735 | 16.3864 | 5.8375 | 0.0009 | **12436.8256** | 16693.3076 | 2024.8256 | **0.0007** |
| | DLinear | 7.6954 | 10.4067 | 13.9395 | 7.9100 | 0.0014 | 20997.7228 | 20952.4161 | 1619.4311 | **0.0007** |
| | FEDFormer | 15.9794 | 16.5863 | 36.0975 | 19.3084 | 0.0015 | 29705.7327 | 55307.0403 | 3245.5183 | 0.0009 |
| | iTransformer | 7.5048 | 10.0161 | **12.7511** | 5.7713 | 0.0010 | 13482.5746 | 18219.6612 | 1591.6404 | 0.0008 |
| | AutoFormer | 7.3876 | 13.9167 | 15.2816 | 7.6855 | 0.0012 | 19628.2867 | 23576.0196 | 1673.8570 | 0.0008 |
| | TimeXer | 8.5213 | 11.4268 | 14.0023 | 5.7325 | 0.0009 | 14397.1884 | 16491.3209 | 1684.9856 | **0.0007** |
| | WPMixer | 6.1543 | 8.9326 | 13.3087 | 5.7842 | 0.0009 | 13205.8932 | 16220.4375 | 1402.8173 | **0.0007** |
| | TimeMixer | 6.0124 | 8.8157 | 13.2158 | 5.7129 | 0.0009 | 14128.4175 | 16645.6821 | 1380.5264 | **0.0007** |
| LLM-based | CrossTimeNet | 8.3125 | 11.6789 | 16.3475 | 6.7924 | 0.0011 | 15120.0853 | 18042.1278 | 1931.2672 | 0.0012 |
| | GPT4TS | 6.9928 | 9.7971 | 15.8238 | 5.7014 | 0.0009 | 13546.1725 | 16839.0718 | 1790.3269 | 0.0010 |
| | TimeLLM | 6.8780 | 9.9814 | 15.8845 | 5.6695 | 0.0010 | 13427.4982 | 16665.2379 | 1575.8937 | 0.0011 |
| | DeepSeek-R1 | 6.7098 | 11.3845 | 14.8561 | 7.0063 | 0.0026 | 29653.1218 | 30780.9011 | 4047.1201 | 0.0021 |
| Ours | Time-R1 | **5.8752** | **8.7093** | 13.1034 | **5.6673** | **0.0007** | 13033.1820 | 16150.5556 | 1353.9381 | **0.0007** |

We implemented the Time-R1 framework on nine datasets. The comparison with baseline models is summarized in Table 2. A more comprehensive list of results for metrics such as MAE can be found in Appendix Table 5. Key observations are as follows:

(1) **Limitations of Traditional Methods and LLM Baselines.** Traditional deep learning-based forecasting models, such as PatchTST, DLinear, and iTransformer, achieve reasonable performance but are limited by their one-step "fast thinking" paradigm, which struggles with complex temporal dependencies and high-level reasoning. LLM-based methods like TimeLLM perform better by leveraging the reasoning abilities of LLMs, especially for long-term and non-linear patterns. However, they still treat forecasting as a direct generation task without explicit step-by-step reasoning, leading to potentially inconsistent or logically flawed predictions. Moreover, their reliance on pre-trained knowledge with minimal task-specific adaptation limits their explainability in the forecasting process.

(2) **Performance Improvement and Advantage of Time-R1.** Our proposed Time-R1 follows a two-stage optimization framework. In the first stage, CoT-guided SFT enables the model to learn structured output formats and basic reasoning logic. In the second stage, we further enhance the model's reasoning capabilities through GRIP with fine-grained reward mechanisms. These include logical consistency, temporal coherence, and multi-horizon accuracy, which iteratively refine the model's reasoning paths. Experimental results show this approach not only improves forecasting performance but also enhances generalization under zero-shot and out-of-distribution settings.

### 5.2 ABLATION STUDY

**Impact of Supervised Fine-tuning and Reinforcement Learning.** Next, we evaluate the necessity of CoT-based SFT by comparing two training strategies: (i) direct RL without SFT, and (ii) SFT

Table 3: Ablation study on Training Strategies, Reward Design, and Template Components.

| Method | | ETTh1 | | ETTm2 | | Wind | |
|---|---|---|---|---|---|---|---|
| | | MSE | MAE | MSE | MAE | MSE | MAE |
| Full Model | Time-R1 | **5.8752** | **1.2325** | **5.6673** | **1.3771** | **1353.9381** | **15.1095** |
| Training Strategies | *w/o* SFT | 6.3558 | 1.4278 | 6.3673 | 1.4850 | 1632.6491 | 16.7903 |
| | *w/o* RL | 13.2196 | 1.7820 | 12.5940 | 3.7759 | 3424.0485 | 28.1392 |
| Reward | *w/o* Length | 5.8781 | 1.2325 | 6.3210 | 1.4024 | 1358.5169 | 15.4612 |
| | *w/o* MSE | 10.0948 | 1.5614 | 9.7449 | 2.4865 | 2749.2582 | 21.0272 |
| | *w/o* Seasonal Decomposition | 6.0132 | 1.2403 | 6.0220 | 1.3859 | 1462.5454 | 15.8056 |
| | *w/o* Trend Decomposition | 7.4775 | 1.3429 | 6.6881 | 1.4523 | 1766.9750 | 16.1789 |
| | *w/o* Structural Similarity | 7.8558 | 1.4278 | 8.5940 | 1.7759 | 2316.0214 | 22.3412 |
| Template Components | *w/o* Timestamps | 9.9146 | 1.5446 | 8.7454 | 2.4867 | 2816.0214 | 26.3412 |
| | *w/o* domain context | 6.2286 | 1.3989 | 6.3517 | 1.5319 | 1639.4436 | 17.0182 |

followed by RL. As illustrated in Figure 4(b), the model trained without SFT suffers from slower convergence and inferior performance, especially in early training stages. In contrast, initializing RL with a well-aligned SFT model significantly accelerates learning and leads to better final performance. This demonstrates that SFT provides a strong foundation for reasoning path generation, which is then further refined through rule-augmented reinforcement learning.

Furthermore, we conducted an ablation study by completely removing RL (see Table 3). The results demonstrate a significant degradation in TSF performance, with absolute performance drops on the ETT and Wind dataset respectively, highlighting RL's crucial role in optimizing SFT-initialized reasoning paths. This finding indicates that while SFT establishes fundamental reasoning patterns, RL provides indispensable optimization through the following mechanisms: (1) discovering higher-reward reasoning trajectories through exploration, and (2) suppressing plausible-yet-incorrect reasoning paths via reward shaping. RL proves to be a critical factor in achieving SOTA performance.

**Impact of Multi-objective Reward Design.** We analyze the impact of each reward term in RL by training models with partial reward components. As shown in Table 3, removing any term degrades performance, indicating that all contribute to forecasting accuracy. The largest drops occur when MSE or Seasonal-Trend Decomposition rewards are removed, emphasizing the importance of point-wise precision and temporal structure. While Format and Length rewards have smaller effects on metrics, they ensure output consistency and training stability. Structural Similarity reward further enhances structural fidelity, especially for complex sequences. More experiments are in Appendix F.2.

**Impact of Training Template Component.** Finally, we assess how different elements of our structured prompts affect model behavior. We consider two main components: (i) explicit timestamp encoding, and (ii) contextual information such as seasonal period and task constraints. Table 3 show that incorporating these components consistently improves both forecasting accuracy and generalization capability, especially under zero-shot and out-of-distribution scenarios. Models without timestamp information struggle to capture long-range dependencies, while those lacking contextual guidance often produce logically inconsistent outputs.

## 5.3 PERFORMANCE COMPARISON W.R.T DIFFERENT RL OPTIMIZATION

We compare the performance of GRIP using two different sampling strategies — Local Random Sampling, and Cluster-based Random Sampling — against GRPO, a commonly used policy optimization method in reasoning-based reinforcement learning. As illustrated in Figure 4(a), the Cluster-based Random Sampling strategy achieves the highest overall performance, slightly outperforming GRPO. This is attributed to its ability to maintain diversity in trajectory selection by clustering samples based on reward values, which helps preserve potentially informative yet low-reward reasoning paths often ignored by greedy methods. In terms of convergence speed, Cluster-based Sampling also leads, followed by Local Sampling, and finally GRPO, which converges the slowest. Although local Sampling explores a larger search space, it tends to overfit high-reward trajectories early on, leading to relatively poor generalization and suboptimal performance.

## 5.4 PERFORMANCE COMPARISON W.R.T DIFFERENT MODEL TYPES

We analyze the training dynamics of Time-R1 across model types, comparing base and instruct models. Using two Qwen2.5 variants, Qwen2.5-7B-Base and Qwen2.5-7B-Instruct. Figure 4(c) shows the base model converges more slowly and starts from a lower performance level. However, it demonstrates stronger learning potential and eventually achieves slightly better results. This suggests

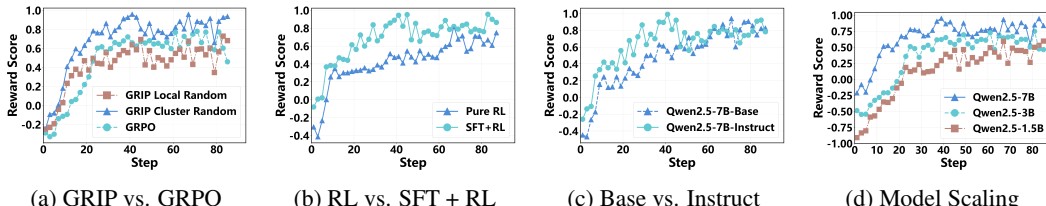

| (a) GRIP vs. GRPO | (b) RL vs. SFT + RL | (c) Base vs. Instruct | (d) Model Scaling |

Figure 4: Figure 2: (a) GRIP vs. GRPO: GRIP converges faster with slightly higher final performance. (b) RL vs. SFT+RL: SFT+RL achieves faster initial convergence and superior final performance. (c) Base vs. Instruct: Instruct model enables faster early reward growth, though base model achieves higher final reward. (d) Model Scaling: Larger models show steeper reward improvement curves.

that while instruction tuning accelerates early learning in time series reasoning, iterative RL-based optimization enables the base model to reach marginally superior performance.

## 5.5 PERFORMANCE COMPARISON W.R.T DIFFERENT MODEL SIZES

To evaluate the scaling behavior of Time-R1, we conduct experiments using models with 1.5B, 3B, and 7B parameters on TSF tasks. As shown in Figure 4(d), forecasting performance consistently improves with increasing model size. The 1.5B model achieves reasonable results on simple datasets but struggles with complex temporal patterns. In contrast, the 3B model demonstrates significantly better accuracy and generalization, while the 7B model achieves the best overall performance, particularly in capturing long-term dependencies and handling out-of-distribution scenarios. These results indicate that larger models can substantially enhance temporal reasoning capabilities.

## 5.6 VISUALIZATION OF THE REASONING PROCESS

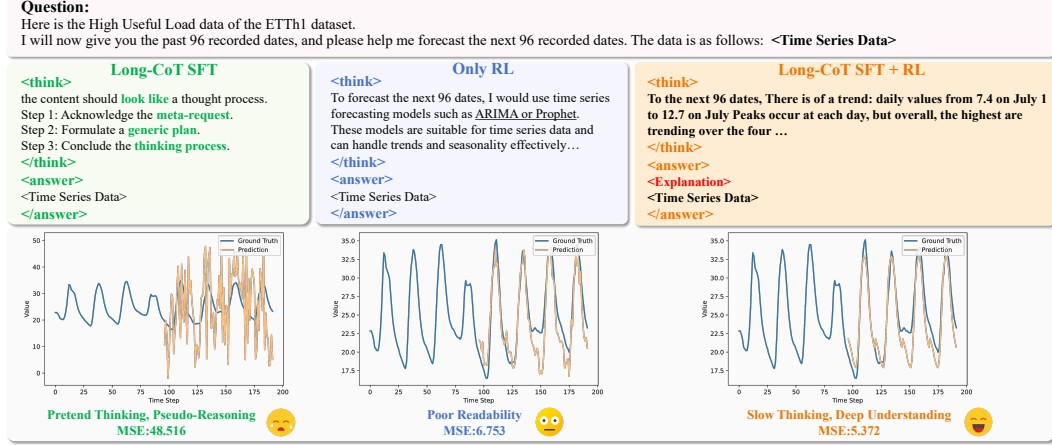

Figure 5: A reasoning case study of long-CoT SFT, RL, and Hybrid Methods on ETTh1 dataset.

As shown in Figure 5, our case study highlights key differences among training paradigms. SFT enables imitation of reasoning patterns but often results in superficial replication, leading to flawed logic and suboptimal performance. Pure RL achieves reasonable accuracy but generates think with poor readability. In contrast, the SFT+RL paradigm not only teaches extended reasoning effectively but, through its reinforcement phase, also improves prediction accuracy while helping the model identify which reasoning components most contribute to performance gains.

## 6 CONCLUSION

In this work, we proposed Time-R1, a generative time series forecasting framework that enables LLMs to perform deliberate reasoning for improved prediction. We introduced time series reasoning by training LLMs to adopt a slow-thinking paradigm, generating explainable intermediate reasoning steps before producing final forecasts, which achieves state-of-the-art TSF performance. Experiments demonstrate that inference time scaling enables substantial improvements in time series reasoning quality, with RL with verified rewards methods yielding stronger generalization to out-of-domain tasks. We hope this work paves the way for future research in structured reasoning.

## 7 ETHICS STATEMENT

We confirm that this work aligns with accepted ethical standards in machine learning research. All data and methodologies used are publicly available or properly cited.

## 8 REPRODUCIBILITY STATEMENT

To support reproducibility, we have provided full details of our experimental setup, including hyperparameters and dataset descriptions, in the experimental section. Code is available.

## 9 THE USE OF LARGE LANGUAGE MODELS (LLMS)

We utilize LLMs to assist and enhance our writing. They help us improve the quality and effectiveness of our textual expression.

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

APPENDIX

# A    RELATED WORK

## A.1    TIMESERIES FORCASTING

Time series forecasting has evolved from classical models like ARIMA, effective under ideal conditions (Box & Jenkins, 1968; Zhang, 2003), to modern deep learning approaches. While ARIMA offers theoretical guarantees, it struggles with real-world data complexities. Machine learning methods (Chen & Guestrin, 2016; Ke et al., 2017) remain highly robust due to their interpretability and ability to model nonlinear relationships. The advent of deep learning introduced sequence-to-sequence models such as Recurrent Neural Networks (RNNs), which initially captured temporal dynamics well (Hewamalage et al., 2021; Salinas et al., 2020). However, RNNs face limitations like restricted receptive fields and error accumulation (Salinas et al., 2020). Advanced architectures incorporating self-attention and convolutional networks have since been developed to capture long-range dependencies (Lai et al., 2018; Li et al., 2019). Concurrently, integrating traditional techniques like trend-seasonal decomposition into neural networks has improved performance (Wen et al., 2020). Notably, even simple linear networks enhanced with decomposition strategies can achieve competitive results (Zeng et al., 2023). Additionally, slice-based methods have shown promise in long-term forecasting by segmenting time series for better accuracy (Nie et al., 2022; Zhang & Yan, 2023). These advancements blend classical principles with deep learning to tackle the challenges of TSF.

## A.2    LLM-BASED TIMESERIES FORCASTING

In recent years, large language models (LLMs) have attracted attention for their ability to understand and generate human-like text, now extending into time series analysis (Zhang et al., 2024; Jiang et al., 2024; Jin et al., 2024). The application of LLMs in this field primarily follows two approaches: fine-tuning and prompt-based zero-shot learning. Fine-tuning involves further training pre-trained LLMs on specific time series data to improve performance (Chang et al., 2023; 2024; Liu et al., 2025b; Jin et al., 2023b), though it requires significant labeled data and computational resources. Conversely, prompt-based zero-shot methods utilize the model's existing knowledge through carefully designed prompts, avoiding task-specific training (Gruver et al., 2023; Liu et al., 2024a). While more flexible and resource-efficient, these methods may not match the performance of fine-tuned models, especially in specialized tasks (Tang et al., 2025; Wang et al., 2024a). Both paradigms illustrate the growing interest in using LLMs for time series, despite challenges in optimizing their effectiveness for such tasks.

## A.3    LARGE LANGUAGE MODELS AND REINFORCEMENT LEARNING

Reinforcement Learning (RL) (Kaelbling et al., 1996) allows an agent to learn decision-making through interactions with its environment, aiming to maximize cumulative rewards. RLHF introduced RL to LLMs via human feedback (Ouyang et al., 2022; Kaufmann et al., 2023), initially training a reward model on human preferences and then using it for tuning the policy LLM, often with Proximal Policy Optimization (PPO). However, PPO's complexity due to multiple optimization rounds poses challenges. To simplify this, methods like Direct Preference Optimization (DPO) (Rafailov et al., 2023) and SimPO (Meng et al., 2024) have been proposed, offering computational efficiency but suffering from off-policy issues (**?**). Another approach, Group Relative Policy Optimization (GRPO) (Shao et al., 2024), avoids a critic model by estimating baselines from group scores, while RLOO (Ahmadian et al., 2024) uses a simplified REINFORCE-style framework. Recently, there has been growing interest in combining CoT reasoning with RL to improve reasoning quality and self-refinement in LLMs (Chu et al., 2025; Chen et al., 2025; Song et al., 2025; Li et al., 2024). Despite these advances, applying RL to enhance LLM-driven reasoning for time series forecasting tasks—particularly through CoT-guided refinement—remains underexplored.

## B   PRELIMINARIES AND ANALYSIS OF RL IN LLMs

### B.1   REINFORCEMENT LEARNING IN LLMs

To leverage Reinforcement Learning (RL) for optimizing Large Language Models (LLMs) in Natural Language Processing (NLP) tasks, the initial and crucial step is to formulate the LLM's generation process as a Markov Decision Process (MDP) Cao et al. (2024); Ouyang et al. (2022). This involves clearly defining the fundamental components of an RL framework: the agent, the environment, states, actions, and rewards. In this context, the LLM itself can be viewed as the agent, interacting with an environment that encompasses the task it is trying to solve (e.g., text generation, question answering). The agent's goal is to learn a policy that maximizes a cumulative reward signal, which reflects how well it performs the given task.

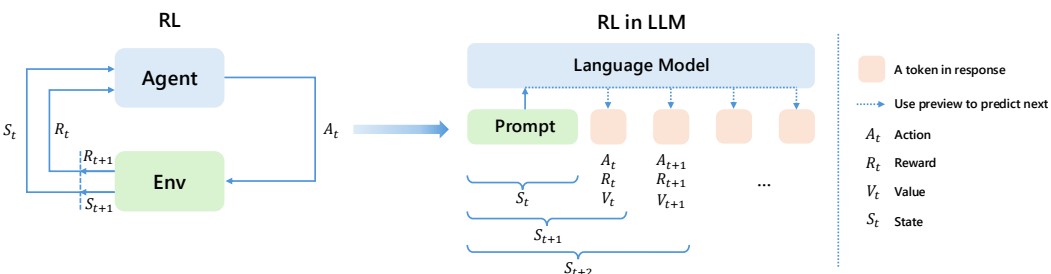

Figure 6: Modeling large language models with reinforcement learning.

As shown in Figure 6, within this MDP framework, the operationalization of RL concepts is specifically tailored to the sequential token-by-token generation characteristic of LLMs. At any discrete time step $t$, the system's current condition is captured by the state $s_t$. This state typically comprises the initial prompt concatenated with the sequence of tokens generated up to that point; thus, the initial state $s_0$ consists solely of the prompt. From a state $s_t$, the LLM executes an action $a_t$, which manifests as the selection and generation of the subsequent token from its predefined vocabulary $V$. The set of all possible tokens constitutes the action space, whose cardinality is denoted as $|V|$. The selection of a particular action $a_t$ is governed by the agent's policy $\pi(a_t|s_t)$, representing a probability distribution over the vocabulary conditional on the current state $s_t$. Following the execution of action $a_t$, the system undergoes a state transition to $s_{t+1}$. In the context of LLM generation, this transition is deterministic, defined by the concatenation $s_{t+1} = [s_t, a_t]$. Crucially for the learning process, a reward signal $r_t$ is provided, and a corresponding value function $V_t$ can be estimated. These elements are contingent upon the specific NLP objective and serve to quantify the desirability of the generated outputs or intermediate actions, thereby guiding the optimization of the LLM's policy.

### B.2   REINFORCEMENT LEARNING WITH VERIFIED REWARD

**Group Relative Policy Optimization.**   The GRPO method Shao et al. (2024) diverges from typical approaches by not requiring a critic model, which often has a comparable size to the policy model. Instead, GRPO calculates the baseline using scores obtained from a group of generated outputs.

Specifically, for every question $q$ sampled from the dataset distribution $P(Q)$, GRPO first employs the existing policy model $\pi_{\theta_{old}}$ to produce $G$ completions, denoted as $\{o_1, o_2, \cdots, o_G\}$. Subsequently, the policy model $\pi_\theta$ is optimized by maximizing a defined objective:

$$\mathcal{J}_{GRPO}(\theta) = \mathbb{E}_{q \sim P(Q), \{o_i\}_{i=1}^{G} \sim \pi_{\theta_{old}}(o|q)} \left\{ \frac{1}{G} \sum_{i=1}^{G} \frac{1}{|o_i|} \sum_{t=1}^{|o_i|} \left\{ \min \left[ \frac{\pi_\theta(o_{i,t}|q, o_{i,<t})}{\pi_{\theta_{old}}(o_{i,t}|q, o_{i,<t})} A_i, \right. \right. \right.$$

$$\left. \left. \left. \text{clip}\left( \frac{\pi_\theta(o_{i,t}|q, o_{i,<t})}{\pi_{\theta_{old}}(o_{i,t}|q, o_{i,<t})}, 1 - \epsilon, 1 + \epsilon \right) A_i \right] - \beta \mathbb{D}_{KL}\left[\pi_\theta || \pi_{ref}\right] \right\} \right\}, \quad (9)$$

$$\mathbb{D}_{KL}\left[\pi_\theta || \pi_{ref}\right] = \frac{\pi_{ref}(o_{i,t}|q, o_{i,<t})}{\pi_\theta(o_{i,t}|q, o_{i,<t})} - \log \frac{\pi_{ref}(o_{i,t}|q, o_{i,<t})}{\pi_\theta(o_{i,t}|q, o_{i,<t})} - 1, \quad (10)$$

In this formulation, $\epsilon$ and $\beta$ function as hyperparameters. The reference model is denoted by $\pi_{\theta_{\mathrm{ref}}}$, typically representing the model's initial state before reinforcement learning is applied. Furthermore, $A_i$ signifies the advantage, calculated from a set of rewards $\{r_1, r_2, \ldots, r_G\}$ that correspond to the various completions within each group:

$$A_i = \frac{r_i - \mathrm{mean}(\{r_1, r_2, \ldots, r_G\})}{\mathrm{std}(\{r_1, r_2, \ldots, r_G\})}. \tag{11}$$

**Rule-based Reward Function.**    Rather than relying on an auxiliary trained reward model, GRPO employs a rule-based system for reward computation. This system calculates the total reward $r_i$ for a given output $o_i$ by aggregating two distinct components, as formalized in Equation 12:

$$2r_i = R_{\mathrm{format}}(o_i) + R_{\mathrm{accuracy}}(o_i), \tag{12}$$

Here, the first component, the format reward $R_{\mathrm{format}}(o_i)$, serves to ensure that the output adheres to the required structural specifications. The second component, the accuracy reward $R_{\mathrm{accuracy}}(o_i)$, is designed to assign substantially higher values to responses that are correct and precise.

## B.3    ANALYSIS OF GRPO

**Analysing Training Cost and Performance Trade-off.**    Equation 9 reveals a linear relationship between GRPO's training overhead and the number of sampled completions. This is primarily due to the necessity of computing probability distributions over all completions for the policy, reference, and old policy models. Taking the DeepSeek-Math experiment as an example, sampling 16 completions per question requires 48 forward passes (16 × 3), leading to a sharp increase in computational cost. Experimental results (Figure 7) show that increasing the number of completions can improve model MSE and MAE on the ETTh1 dataset, with diminishing returns in performance gains. More critically, reducing the number of completions to lower computational load significantly degrades the reasoning capability of the Qwen2.5-7B-Instruct model, highlighting the limitations of conventional approaches.

**Analyzing Optimization Opportunities from Contribution Heterogeneity.**    Recent work Lin et al. (2025) reveals substantial heterogeneity in the contribution of individual completions to training effectiveness. A small subset of high-value samples can provide optimization signals up to tens of times stronger than ordinary ones. This non-uniform distribution opens new avenues for improving training efficiency: by dynamically identifying and prioritizing high-contribution samples, it becomes possible to reduce computational overhead to as low as one-third or even one-fifth of the original level while maintaining model performance (see Section C for detailed optimization strategies). These findings not only explain the efficiency bottlenecks in existing methods but also lay a theoretical foundation for the design of adaptive sampling strategies in future work.

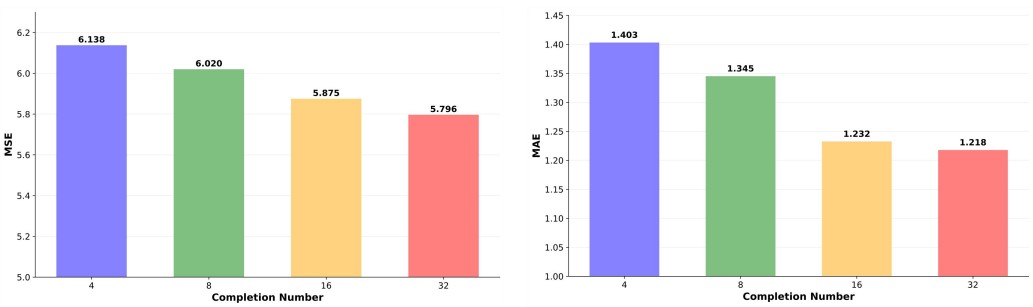

Figure 7: Completion number *vs.* (left) MSE ↓ and (right) MAE ↓. Experiments are conducted on ETTh1 using Qwen2.5B-7B-Instruct.

**Analysising Completion Contribution.** To evaluate the contribution of each completion to the training of the policy model, we analyze the derivative structure of the objective function in Eq. (9) with respect to the model parameters $\theta$ as:

$$\nabla_\theta J_{GRPO}(\theta) = \mathbb{E}_{[q\sim P(Q),\{o_i\}_{i=1}^G\sim\pi_{\theta_{old}}(O|q)]} \left\{ \frac{1}{G}\sum_{i=1}^G \frac{1}{|o_i|}\sum_{t=1}^{|o_i|}\left[ \nabla_\theta\left(\frac{\pi_\theta\left(o_{i,t}|q,o_{i,<t}\right)}{\pi_{\theta_{old}}\left(o_{i,t}|q,o_{i,<t}\right)}A_i\right) \right.\right.$$

$$\left.\left. -\beta\left(\nabla_\theta\frac{\pi_{ref}\left(o_{i,t}|q,o_{i,<t}\right)}{\pi_\theta\left(o_{i,t}|q,o_{i,<t}\right)} - \nabla_\theta\log\frac{\pi_{ref}\left(o_{i,t}|q,o_{i,<t}\right)}{\pi_\theta\left(o_{i,t}|q,o_{i,<t}\right)}\right) \right]\right\}$$

$$=\mathbb{E}_{[q\sim P(Q),\{o_i\}_{i=1}^G\sim\pi_{\theta_{old}}(O|q)]} \left\{ \frac{1}{G}\sum_{i=1}^G \frac{1}{|o_i|}\sum_{t=1}^{|o_i|}\left[ \frac{\nabla_\theta\pi_\theta\left(o_{i,t}|q,o_{i,<t}\right)}{\pi_{\theta_{old}(o_{i,t}|q,o_{i,<t})}}A_i \right.\right.$$

$$\left.\left. +\beta\left(\frac{\pi_{ref}\left(o_{i,t}|q,o_{i,<t}\right)\nabla_\theta\pi_\theta\left(o_{i,t}|q,o_{i,<t}\right)}{\pi_\theta^2\left(o_{i,t}|q,o_{i,<t}\right)} - \frac{\nabla_\theta\pi_\theta\left(o_{i,t}|q,o_{i,<t}\right)}{\pi_\theta\left(o_{i,t}|q,o_{i,<t}\right)}\right) \right]\right\}$$

$$=\mathbb{E}_{[q\sim P(Q),\{o_i\}_{i=1}^G\sim\pi_{\theta_{old}}(O|q)]} \left\{ \frac{1}{G}\sum_{i=1}^G \frac{1}{|o_i|}\sum_{t=1}^{|o_i|}\left[ \frac{\pi_\theta\left(o_{i,t}|q,o_{i,<t}\right)}{\pi_{\theta_{old}}\left(o_{i,t}|q,o_{i,<t}\right)}A_i \right.\right.$$

$$\left.\left. +\beta\left(\frac{\pi_{ref}\left(o_{i,t}|q,o_{i,<t}\right)}{\pi_\theta\left(o_{i,t}|q,o_{i,<t}\right)} - 1\right) \right]\frac{\nabla_\theta\pi_\theta\left(o_{i,t}|q,o_{i,<t}\right)}{\pi_\theta\left(o_{i,t}|q,o_{i,<t}\right)}\right\}$$

$$=\mathbb{E}_{[q\sim P(Q),\{o_i\}_{i=1}^G\sim\pi_{\theta_{old}}(O|q)]} \left\{ \frac{1}{G}\sum_{i=1}^G \frac{1}{|o_i|}\sum_{t=1}^{|o_i|}\left[ \underbrace{\frac{\pi_\theta\left(o_{i,t}|q,o_{i,<t}\right)}{\pi_{\theta_{old}}\left(o_{i,t}|q,o_{i,<t}\right)}}_{Probability\ ratio}\underbrace{A_i}_{Advantage} \right.\right.$$

$$\left.\left. +\underbrace{\beta\left(\frac{\pi_{ref}\left(o_{i,t}|q,o_{i,<t}\right)}{\pi_\theta\left(o_{i,t}|q,o_{i,<t}\right)} - 1\right)}_{KL\ divergence\ constraint} \right]\underbrace{\nabla_\theta\log\pi_\theta\left(o_{i,t}|q,o_{i,<t}\right)}_{Policy\ model\ gradient}\right\}.$$

$$(13)$$

This analysis reveals four key factors influencing policy updates:

1. *Advantage*, which assesses the value of a completion in improving expected returns through the advantage function. A higher advantage indicates stronger reward alignment, making the completion more influential in guiding policy updates.

2. *Probability ratio*, wich compares the likelihood of an action under the current policy $\pi_\theta$ to that under the old policy $\pi_{\theta_{old}}$. It amplifies actions favored by the new policy and suppresses those preferred by the old one, guiding the policy toward higher rewards. A higher ratio signifies greater confidence in the action, influencing the optimization process more significantly. This term is crucial for identifying high-value completions when combined with the advantage function.

3. *KL divergence*, which measures the deviation of the current policy from the reference model. It enforces stability during training by penalizing excessive changes, but does not directly contribute to reasoning pattern formation.

4. *Policy model gradient*, which indicates the direction of parameter updates.

Previous research Hu et al. (2025) has shown that removing the KL divergence constraint does not significantly affect the model's reasoning performance, as the core learning signal primarily comes from the advantage term aligned with rewards. Furthermore, we decompose the core expression for policy updates into two components: the *probability ratio* and the *advantage* term. For a completion to make a significant contribution to training, both components must have substantial values. If either of them is close to zero, the overall contribution will also be negligible. By removing the KL divergence term and decoupling its regularization effect, we derive the simplified form of GRPO's objective derivative as:

$$\nabla_\theta J_{GRPO}(\theta) \approx \mathbb{E}_{\left[q \sim P(Q), \{o_i\}_{i=1}^G \sim \pi_{\theta_{old}}(O|q)\right]}$$

$$\left\{ \frac{1}{G} \sum_{i=1}^{G} \frac{1}{|o_i|} \sum_{t=1}^{|o_i|} \left[ \underbrace{\frac{\pi_\theta\left(o_{i,t}|q, o_{i,<t}\right)}{\pi_{\theta_{old}}\left(o_{i,t}|q, o_{i,<t}\right)}}_{\text{Probability ratio}} \cdot \underbrace{A_i}_{\text{Advantage}} \right] \underbrace{\nabla_\theta \log \pi_\theta\left(o_{i,t}|q, o_{i,<t}\right)}_{\text{Policy model gradient}} \right\}, \quad (14)$$

This simplified formulation focuses on the reward-driven learning signal while preserving the essential gradient dynamics required for effective policy optimization. With the help of this simplified expression, we can evaluate the advantage value of each completion before the full model computation is carried out. Specifically, if a completion exhibits a very small absolute advantage value, its contribution to the policy update is negligible. We can filter out such low-value completions at an early stage, thereby avoiding redundant forward and backward computations.

To further improve training efficiency and learning effectiveness, we could introduce a sampling strategy based on reward values or advantage estimates. Unlike uniform sampling across all completions, we prioritize those with higher advantage values for inclusion in the training process. These samples typically contain stronger learning signals and are more valuable for policy updates. By increasing the sampling probability of these high-impact completions during batch construction—or through upsampling techniques—we not only reduce computational overhead but also significantly accelerate convergence and enhance final model performance.

This approach ensures efficient training iterations while maintaining the quality of policy updates, achieving a favorable balance between computational cost and learning effectiveness. As a result, it enables a dual improvement in both training efficiency and model capability.

## C  GROUP-BASED RELATIVE IMPORTANCE FOR POLICY OPTIMIZATION

We introduce GRIP (Group-based Relative Importance for Policy Optimization), a general RL optimization method designed for optimizing entire trajectories generated by LLMs as time series forecasting reasoners. Unlike GRPO, which linearly increases inference cost with the number of completions due to uniform sampling and equal weighting, GRIP adopts a non-uniform sampling strategy that selects a small subset of high-reward trajectories from a larger candidate pool, significantly reducing forward passes. Furthermore, GRIP employs adaptive weighting to amplify gradient signals from informative samples, enabling more efficient learning from high-value completions. This design not only reduces compute burden but also mitigates the diminishing returns of increased sample count, thereby enhancing both training efficiency and forecasting accuracy. The GRIP objective function, formally defined in Equation 15, operates within a policy gradient framework and integrates a non-uniform sampling strategy with adaptive trajectory weighting.

$$\mathcal{J}_{\text{GRIP}}(\theta) = \mathbb{E}_{\substack{q \sim P(Q), \\ \{o_j\}_{j=1}^{k \cdot G} \sim \pi_{\theta_{\text{old}}}(o|q), \\ \{o_i\}_{i=1}^{G} \sim \text{Sample}\left(\{o_j\}_{j=1}^{k \cdot G}; R(o_j)\right)}} \left\{ \sum_{i=1}^{G} w_i^U \frac{1}{|o_i|} \left\{ \sum_{t=1}^{|o_i|} \min\left[ \frac{\pi_\theta(o_{i,t}|q, o_{i,<t})}{\pi_{\theta_{\text{old}}}(o_{i,t}|q, o_{i,<t})} A_i, \right. \right. \right.$$

$$\left. \left. \left. \text{clip}\left( \frac{\pi_\theta(o_{i,t}|q, o_{i,<t})}{\pi_{\theta_{\text{old}}}(o_{i,t}|q, o_{i,<t})}, 1-\epsilon, 1+\epsilon \right) A_i \right] - \beta \mathbb{D}_{KL}[\pi_\theta || \pi_{ref}] \right\} \right\}. \quad (15)$$

where $\epsilon$ and $\beta$ are hyperparameters. $\pi_{\text{ref}}$ is the reference model, typically initialized as the pre-trained model before reinforcement learning begins. The output $\{o_i\}$ is selected through a sampling process from policy $\pi_{\theta_{\text{old}}}$. The hyperparameter $k$ controls the size of the rollout space, while $G$ referred to as the group size. $\mathbb{D}_{KL}$ represents the KL divergence, which is incorporated into the loss function as a regularization term during training. And $A_i$ is the advantage computed using a group of rewards $\{r_1, r_2, \ldots, r_G\}$ corresponding to the completion trajectories within each group. The weight $w_i^U$ denotes the adaptive weighting assigned to each trajectory. This objective balances exploration and exploitation while mitigating gradient dilution. The sampling strategy and adaptive weight will be discussed in the following section.

**GRIP Pipeline.** To elucidate its operational mechanics, the GRIP algorithm is implemented through a structured pipeline. The distinct stages of this process are outlined as follows:

(1) The old policy model samples $k$ groups of $G$ completions for each question, a total of $k \cdot G$.

(2) The reward function computes the reward for each completion (Sec. 3.4.1).

(3) GRIP non-uniformly samples $G$ completions based on rewards to balance exploration and exploitation (Sec. 3.4.2).

(4) The advantage of each completion is calculated, and adaptive weighting is employed to assign greater significance to high-quality reasoning paths among these completions (Sec. 3.4.2).

(5) Subsequently, the policy model is updated, with its gradient signals effectively formed as a weighted average of the selected completions, reflecting this assigned path-dependent significance.

GRIP differs significantly from GRPO in both its initial sampling strategy and the mechanism for weighting completion contributions during policy updates. GRPO typically generates $G$ completions directly and employs an arithmetic mean when processing their outcomes. In contrast, GRIP first explores a broader set by sampling $k \cdot G$ completions, from which $G$ are subsequently selected for the update phase. Critically, while GRPO's use of an arithmetic mean implies equal consideration for its $G$ samples, GRIP's policy update leverages an adaptive weighted average. This ensures that high-quality reasoning paths within the selected $G$ completions exert a more substantial influence on the gradient signals, thereby fostering more robust and effective learning.

## D  CoT-based Supervised Fine-tuning Data collection

---

**Algorithm 1** Three-stage CoT data construction for Ours

---

1: **Input:** $\mathcal{T}$ := Set of training time series samples with historical data and ground truth labels
2: **Output:** $\mathcal{D}_{\text{SFT}}$ := Structured Chain-of-Thought dataset for SFT
3: $\mathcal{D}_{\text{SFT}} \leftarrow \emptyset$                                                *Initialize the output dataset*
4: **for** $t \in \mathcal{T}$ **do**
5:    $x_{\text{hist}} \leftarrow \text{ExtractHistorical}(t)$
6:                                        *Extract historical time series input*
7:    $y_{\text{pred}}^{(1)}, \ldots, y_{\text{pred}}^{(k)} \leftarrow \text{DeepSeek-R1}(x_{\text{hist}}; \text{strict formatting})$
8:                                      *Generate $k$ candidate predictions*
9:    $\text{MAPE}_i \leftarrow \text{ComputeMAPE}(y_{\text{pred}}^{(i)}, y_{\text{true}})$ for each candidate
10:                                               *Evaluate using MAPE*
11:    $y^* \leftarrow \text{SelectMinMAPE}(\{y_{\text{pred}}^{(i)}\})$
12:                                                  *Select best prediction*
13:    $\text{CoT}^{(1)}, \ldots, \text{CoT}^{(k)} \leftarrow \text{DeepSeek-R1}(\text{prompt} = x_{\text{hist}}, y_{\text{true}}, \text{CoT of } y^*)$
14:                        *Prompt to generate reasoning paths based on ground truth label*
15:    $\text{CoT}^* \leftarrow \text{SelectHighQuality}(\{\text{CoT}^{(i)}\}, y_{\text{true}})$
16:                             *Select CoT aligned with ground truth label*
17:    $\text{sample} \leftarrow \text{Concatenate}(\text{<think>}, \text{CoT}^*, \text{<\think>}, \text{<answer>}, y_{\text{true}}, \text{<\answer>})$
18:                                 *Combine reasoning and true answer*
19:    $\mathcal{D}_{\text{SFT}} \leftarrow \mathcal{D}_{\text{SFT}} \cup \{\text{sample}\}$
20:                                                *Add to final dataset*
21: **end for**

---

## E  Data Statistics

We presents the statistical characteristics of our datasets, shown in table 4.

Table 4: Statistics information of experimental datasets.

| Dataset | Domain | Timestamps | Features | Frequency |
|---|---|---|---|---|
| ETTh1&ETTh2 | Electricity | 17,420 | 7 | 1 hour |
| ETTm1&ETTm2 | Electricity | 69,680 | 7 | 15 mins |
| AQWan&AQShunyi | Environment | 35,064 | 11 | 1 hour |
| Exchange | Economy | 7,588 | 8 | 1 day |
| Wind | Energy | 48,673 | 7 | 15 mins |
| NASDAQ | Stock | 1,244 | 5 | 1 day |

# F ADDITIONAL RESULT AND ANALYSIS

## F.1 MAIN RESULTS

Due to page limitations, we present the main results using the MAE evaluation metric here in Table 5. Compared to both traditional methods and LLM-based approaches, Time-R1 achieves competitive improvements.

Table 5: Performance comparison of Time-R1 and baseline models with best values in bold and second-best underlined. MAE ↓ is used as the evaluation metric. Deepseek-R1 denotes zero-shot using our reasoning template. Overall, Time-R1 achieves superior performance across dataset.

| | Methods | ETTh1 | ETTh2 | ETTm1 | ETTm2 | Exchange | AQWan | AQShunyi | Wind | NASDAQ |
|---|---|---|---|---|---|---|---|---|---|---|
| Traditional | PatchTST | 1.6376 | 2.0076 | 1.9446 | 1.3973 | 0.0200 | 39.5089 | 42.5427 | 20.4654 | **0.0213** |
| | DLinear | 1.5116 | 2.0855 | 1.7815 | 1.9136 | 0.0256 | 51.7896 | 50.0134 | 18.6036 | 0.0217 |
| | FEDFormer | 2.2584 | 2.3487 | 2.5178 | 2.7729 | 0.0247 | 56.3129 | 50.3959 | 24.2476 | 0.0218 |
| | iTransformer | 1.5126 | 1.9295 | 1.6609 | 1.4598 | 0.0204 | 40.3070 | 42.5116 | 18.1864 | 0.0237 |
| | AutoFormer | 1.5945 | 2.3246 | 2.0989 | 1.7696 | 0.0230 | 48.4358 | 51.4833 | 19.6086 | 0.0229 |
| | TimeXer | 1.5596 | 2.0895 | 1.7785 | 1.4419 | 0.0195 | 41.8280 | 41.4929 | 18.4876 | 0.0233 |
| | WPMixer | 1.4321 | 2.0013 | 1.7502 | 1.4205 | 0.0188 | 41.2347 | 40.9811 | 18.1034 | 0.0225 |
| | TimeMixer | 1.4018 | 2.0456 | 1.7103 | 1.3987 | 0.0182 | 40.6789 | **40.3452** | 18.8801 | 0.0219 |
| LLM-based | CrossTimeNet | 1.5492 | 2.1731 | 1.9863 | 1.5281 | 0.0217 | 43.6267 | 46.0941 | 20.6205 | 0.0252 |
| | GPT4TS | 1.4299 | 1.9134 | 1.9635 | 1.4943 | 0.0202 | 40.2815 | 42.3969 | 18.4919 | 0.0256 |
| | TimeLLM | 1.4286 | 1.9143 | 1.9626 | 1.4926 | 0.0201 | 40.2746 | 42.3926 | 18.4890 | 0.0255 |
| | DeepSeek-R1 | 1.4316 | 2.0165 | 1.8876 | 1.6289 | 0.0209 | 64.9627 | 57.6412 | 30.6469 | 0.0231 |
| Ours | Time-R1 | **1.2325** | **1.7192** | **1.6510** | **1.3771** | **0.0161** | **39.1846** | 46.5032 | **15.1095** | 0.0217 |

## F.2 ANALYSIS ABOUT REWARD DESIGN

**Distance Metric in Accuracy Reward.** To investigate the impact of different distance metrics on the accuracy reward during the reinforcement learning phase, we conducted experiments on multiple real-world time series forecasting tasks using the Exchange, AQShunyi, and NASDAQ datasets, shown in Figure 8. Specifically, we evaluated Mean Squared Error (MSE), Mean Absolute Error (MAE), Dynamic Time Warping (DTW), and Mean Absolute Percentage Error (MAPE) as accuracy reward signals within our GRIP optimization framework. The experimental results demonstrate that MSE consistently outperformed other metrics, followed by MAE, while DTW and MAPE showed relatively limited improvements. MSE penalizes the squared prediction errors, making it more sensitive to outliers and thereby guiding the model to focus on overall consistency between predictions and ground truth values. This leads to enhanced stability in multi-step forecasts. In contrast, although MAE is more robust due to its linear response to errors, it suffers from less concentrated gradient updates, affecting convergence efficiency. DTW, despite its ability to handle temporal misalignment, introduces computational complexity and asymmetry issues, making it challenging to integrate effectively into end-to-end training. Additionally, MAPE can suffer from numerical instability when target values are close to zero, limiting its practical utility in training. In conclusion, we recommend prioritizing MSE as the primary accuracy reward metric during the reinforcement learning optimization phase to achieve superior predictive accuracy and reasoning stability. Auxiliary reward terms may be incorporated based on specific task requirements to further enhance model robustness.

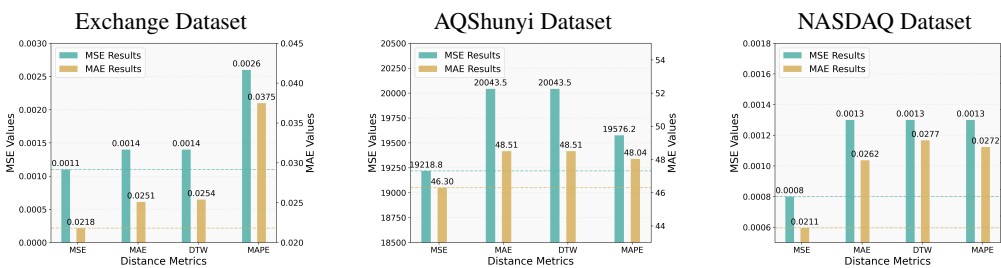

Figure 8: Experimental Results Display Base on MSE, MAE, DTW, MAPE Distance Metrics.

Table 6: Time series forecasting results of Time-R1 comparing with SOTA baseline TimeLLM across four datasets. All results are averaged from 3 different predicted window of {96, 192, 336}.

| Method | Predicted window | Time-R1 | | TimeLLM | |
|---|---|---|---|---|---|
| | | MSE | MAE | MSE | MAE |
| ETTh1 | 96 | **5.8752** | **1.2325** | 6.8780 | 1.4286 |
| | 192 | **6.4146** | **1.2693** | 7.7990 | 1.4989 |
| | 336 | **6.6886** | **1.3288** | 9.3575 | 1.5617 |
| ETTh2 | 96 | **8.7093** | **1.7192** | 9.9814 | 1.9143 |
| | 192 | **9.4605** | **1.8868** | 10.4066 | 2.0755 |
| | 336 | **14.3501** | **2.2914** | 15.7851 | 2.6351 |
| ETTm1 | 96 | **13.1034** | **1.6510** | 15.8845 | 1.9626 |
| | 192 | **15.0766** | **2.2640** | 16.5843 | 2.2904 |
| | 336 | **18.0009** | **2.3299** | 19.8010 | 2.5629 |
| ETTm2 | 96 | **5.6673** | **1.3771** | 5.6695 | 1.4926 |
| | 192 | **6.8172** | **1.5331** | 7.1581 | **1.5264** |
| | 336 | **8.5726** | 1.7476 | 9.0012 | **1.7224** |

## F.3 IMPACT OF PREDICTED WINDOW LENGTH

The results in Table 6 demonstrate that Time-R1 consistently outperforms TimeLLM across multiple datasets and prediction horizons, particularly in long-sequence rolling forecasting tasks. Our approach employs a rolling prediction strategy: first using the historical 96 time steps to predict the subsequent 96 steps, and then recursively feeding the predicted values back as input to forecast further into the future. In the ETTh1 dataset, Time-R1 achieves an MSE of 6.4146 and an MAE of 1.2693 under the 192-step forecasting window, substantially lower than TimeLLM's MSE of 7.7990 and MAE of 1.4989, highlighting its superior capability in multi-step rolling prediction. Under the longer 336-step forecasting window, Time-R1 obtains an MSE of 6.6886, still significantly outperforming TimeLLM's MSE of 9.3575, which indicates that the model maintains high accuracy and stability even after multiple recursive prediction steps.

## F.4 CASE STUDY: AN INPUT AND OUTPUT EXAMPLE OF TIME-R1

We provide a complete input-output example of Time-R1 for reference, with some time steps omitted, shown in Table 8.

## F.5 VISUALIZATION OF PREDICTION RESULTS

In this visualization (Figure 9), Time-R1 is compared against six baseline methods, including LLM-based approaches (GPT4TS and TimeLLM) and traditional models (PatchTST, iTransformer, Autoformer, and TimeXer). Time-R1 consistently demonstrates more accurate and smoother predictions, closely aligning with the ground truth. In contrast, the version of Time-R1 trained with only SFT performs poorly, yielding subpar forecasting results. On the other hand, Time-R1 with only RL achieves improved performance, outperforming the baselines to some extent.

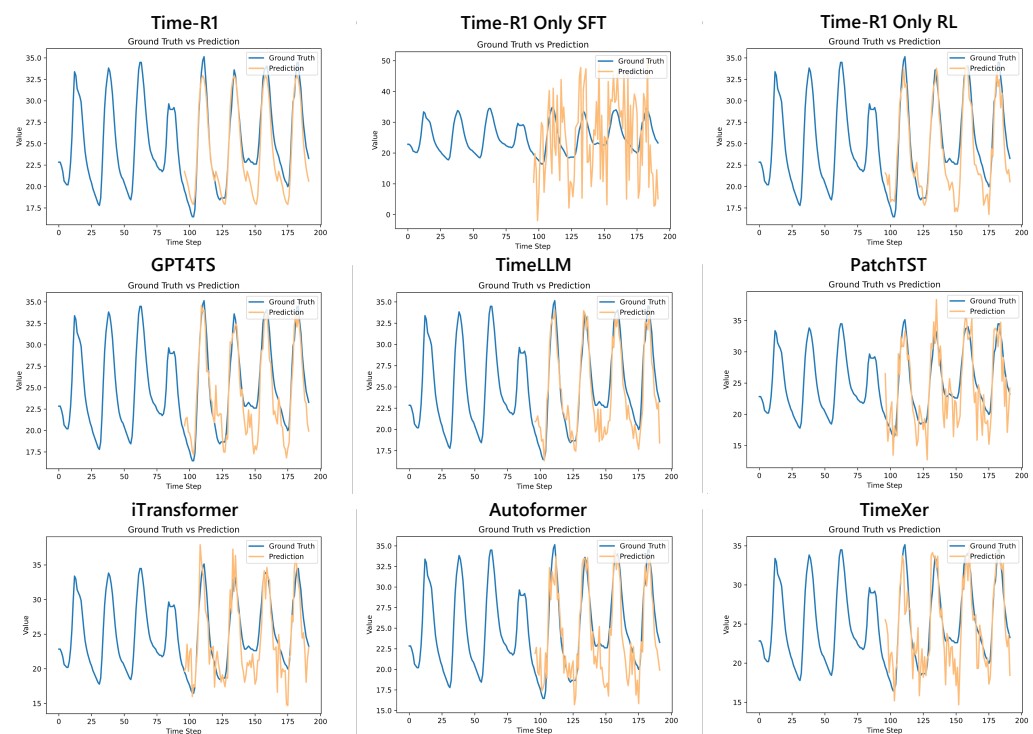

Figure 9: Illustration of forecasting showcases comparing Time-R1 and baseline models. The look-back window and the predicted window are set to 96 for the ETTh2 dataset.

Table 7: Comprehensive comparison of Time-R1 and PatchTST in terms of inference efficiency, training resources, training duration, and generalization.

| Method | Inference Speed (samples/s) | GPU Memory (MB) | Training Cost | Training Duration | Generalization |
|---|---|---|---|---|---|
| PatchTST | 790.30 | 583 | 1x RTX 4090 (24GB) | 10 minutes | No (per-dataset training) |
| Time-R1 (Ours) | 2.93 | 22,502 | 4x A800 (80GB each) | 8 hours | Yes (train once, deploy across datasets) |

## F.6   TRADE-OFF BETWEEN ACCURACY AND INFERENCE EFFICIENCY

While the gains in Table 1 appear modest, they stem from a paradigm prioritizing reasoning depth, interpretability, and generalization over speed. As shown in Table 7, Time-R1 (2.93 samples/s, 22,502 MB) incurs high cost due to its LLM-based multi-step reasoning, unlike the lightweight PatchTST (790.30 samples/s, 583 MB). This "slow-thinking" design is intentional, targeting high-impact, latency-tolerant applications—such as financial, energy, or environmental forecasting—where accuracy and trust matter more than real-time response. Time-R1 offers interpretable reasoning, strong generalization, and a foundation for future improvements, making it suitable for scenarios where decisions are costly and explanations are essential.

## G   LIMITATIONS

Despite significant advancements in enhancing the model's ability to capture high-frequency temporal patterns and improve time series reasoning, challenges remain due to computational limitations that prevent experimentation with larger open-source models. Additionally, extending the length of the prediction window is not feasible. Consequently, our method may encounter inaccurate predictions when dealing with complex temporal structures or long-term forecasting requirements.

## H   BROADER IMPACT

This paper presents work whose goal is to advance the field of Machine Learning. There are many potential societal consequences of our work, none which we feel must be specifically highlighted.

Table 8: Case study of complete input-output example of Time-R1.

**Input:**

Here is the High UseFul Load data of the transformer.I will now give you data for the past 96 recorded dates, and please help me forecast the data for next 96 recorded dates. The data is as follows:

```
date                    HUFL
2016-07-01 00:00:00     5.827
2016-07-01 01:00:00     5.693
2016-07-01 02:00:00     5.157
2016-07-01 03:00:00     5.090
2016-07-01 04:00:00     5.358
2016-07-01 05:00:00     5.626
2016-07-01 06:00:00     7.167
2016-07-01 07:00:00     7.435
2016-07-01 08:00:00     5.559
2016-07-01 09:00:00     4.555
. . .''
```

Please give me the complete data for the next 96 recorded dates, remember to give me the complete data. You must first conduct reasoning inside <think> ...</think>. When you have the final answer, you can output the answer inside <answer>...</answer>.

**Output:**

<think>Summary of HUFL 96-Hour Forecast Reasoning.

1. Basic Patterns and Trends in the Data: The HUFL dataset spans 96 hours from July 1 to July 4, 2016, with values ranging from 4.2 to 12.7. Daily maximums increase from 7.4 to 12.7 over four days, showing an upward trend despite varying peak times.

2. Observed Seasonality or Daily Fluctuations: Daily patterns show lower values at night (00:00–5:00) and higher values during mornings and midday. Morning peaks occur between 6–8 AM, midday peaks around 11 AM–noon, both increasing daily. Evening rises (6–10 PM) also show growth but with some irregularity in peak timing and magnitude.

3. Forecasting Methods and Assumptions Made: Forecasting assumes daily seasonality and a slowing upward trend based on observed data. Using July 4th as a base, each subsequent day's hourly values are incremented by small amounts (0.1–0.3 or 2–5%), more for peak hours. Trend may plateau after recent highs; alternative patterns like reversals were considered but not confirmed due to limited data.

4. Potential Limitations or Uncertainties: Only four days of data limit identification of long-term trends or weekly cycles. Trend continuation is speculative, external factors are unknown, and inherent variability reduces forecast accuracy.

Conclusion: The forecast extends observed daily seasonality and recent upward trends, incrementally adjusting July 4th's values. Despite maintaining daily fluctuation structures, the short data window and lack of validation mean the forecast is an informed estimate. </think>
<answer>
```
date                    attr
2016-07-05 00:00:00     11.989
2016-07-05 01:00:00     12.525
2016-07-05 02:00:00     12.324
2016-07-05 03:00:00     10.717
2016-07-05 04:00:00     11.320
2016-07-05 05:00:00     10.851
2016-07-05 06:00:00     13.329
2016-07-05 07:00:00     11.454
2016-07-05 08:00:00     11.052
2016-07-05 09:00:00     10.985
... ''
```
</answer>

