# OpenReview forum: "Time Series Forecasting as Reasoning: A Slow-Thinking Approach with Reinforced LLMs"
_ICLR.cc/2026/Conference — Submitted to ICLR 2026_

### Official Review · Reviewer_nXNz · 2025-10-17

**Soundness:** 2
**Presentation:** 3
**Contribution:** 2
**Rating:** 2
**Confidence:** 4

**Summary:**

The paper introduces Time-R1, a two-stage reinforcement fine-tuning framework designed to enhance large language models (LLMs) for time series forecasting (TSF) by developing explicit multi-step reasoning capabilities.

**Strengths:**

- Several LLM-based models are used as benchmarks. Competitive forecasting benchmarks like PatchTST are also included.
- Presentation and results in Table 3 are clear and well-structured.
- The purposed approach appears to outperform most baselines across datasets.
- The figures are visually effective and contribute to understanding the proposed framework.
- The paper’s proposed framework for fine-tuning dataset construction is a valuable direction.

**Weaknesses:**

Several claims in the paper motivating use of reasoning for time series models are either vague or overstated:
   - For example, the statement “These models map history to future directly without detecting regime changes or performing step-by-step inference, resembling fast (not deliberate) thinking for time series” lacks precision. The notion of “regime changes” is not well defined. Modeling covariates could, for instance, already capture temporal dynamics associated with external events.
   - Lines 62–63: The phrase “time series often reflect more complex temporal logic, which should not merely be ‘fitted’—they should be understood and reasoned” is conceptually unclear. What is meant by “complex temporal logic” (e.g., domain knowledge, text-based covariates)? The authors should better justify why “reasoning” about such logic is necessary or beneficial compared to existing data-driven approaches.
   - Line 84: The assertion that the model is “fine-tuned for memorization” is a strong and potentially misleading claim. Forecasting fine-tuning does not inherently imply memorization, and the distinction between memorization and generalization is an active topic of research that requires supporting evidence.

Methodology contribution concerns:
- Reasoning evaluation (Eq. 1): The evaluation procedure for reasoning remains unclear. If the metric is based solely on achieving the lowest MSE, it does not actually assess whether the model engages in reasoning, it merely rewards numerical accuracy.

- Reward design (Eqs. 3 and 4): The proposed reward functions appear conceptually similar to traditional loss formulations such as MSE and trend–seasonality decomposition. This raises questions about how they differ from conventional forecasting objectives. The authors critique prior approaches for lacking explicit reasoning, yet the reward relies on classical statistical constructs. The paper should clarify how this formulation advances the goal of reasoning-based forecasting. Moreover, this seems inconsistent with statements in the introduction (Lines 51–52): “Although effective in benchmarks, their underlying logic is largely based on pattern recognition (Cheng et al., 2025a) and trend prediction, lacking an explicit reasoning process.”

- Model architecture clarity: The underlying architecture on which Time-R1 is built is not clearly described. Rather than presenting Time-R1 as a standalone model, the contributions of the reasoning-oriented fine-tuning and reinforcement learning framework, would be more compelling if shown to consistently improve performance across multiple LLM backbones.

Benchmark concerns:
- The paper lacks comparisons with non-LLM-based Time Series Foundation Models (e.g., TimesFM, Moirai, Moment) and classical statistical baselines (e.g., ARIMA, ETS). Including such comparisons is crucial to validate its claimed advantages and motivate the use of LLMs for forecasting.

**Questions:**

Other:
- Lines 261-269: formulation of the GRIP Objective could be more clearly articulated to outline the provided equations.
- The purposed multi-trajectory approach in lines 292–300 is interesting. It could be interesting to explore ensembling of top-k sample trajectories. Such approach could reduce forecast variance an important consideration for forecasting in practice.
- In addition to the reward based on returned prediction sequence length, including a reward function based on prediction variance from distributional loss functions could help further improve model performance.
- Normalization is very often used in time series forecasting models. It could be interesting to show an ablation study with and without normalization to see how robust this approach is.
- Some acronym and functions are not defined:
   - Function ‘g’ (Line 107) not defined.
   - SFT (Lines 095, 151) not defined.
   - Extrema (Eq. 5) and Eq. 6 unclear. Unclear how extrema are determined.

---

> ### Author Response · Authors · 2025-11-21
> **Official Response to Reviewer nXNz[1/3]**
>
> We sincerely thank the reviewer for an exceptionally thorough and insightful report. We were very positively surprised by how carefully you engaged with the paper and by the number of concrete, actionable suggestions you provided. Your comments greatly help us clarify the framing and presentation of the work and will significantly improve both the methodology description and the empirical evaluation of Time-R1. Below we provide our responses to your questions and suggestions.
>
> > **W1 & 2: Claims regarding "regime changes" and "complex temporal logic" are vague. Justification is needed for why reasoning is superior to existing data-driven approaches.**
>
> You are absolutely right that precise definitions are crucial here. By "regime changes," we refer to **non-stationary distribution shifts** (e.g., sudden market volatility or sensor drift) that purely statistical models often struggle to adapt to dynamically. By "complex temporal logic," we refer to causal or conditional dependencies (e.g., "if specific seasonal peaks occur in the morning, the evening trend usually flattens") that require multi-step inference rather than simple point-to-point mapping.
> Existing data-driven approaches often act as "black boxes" that implicitly mix these factors in latent space. In contrast, our "reasoning" approach forces the model to explicitly decompose these steps (Trend Analysis $\to$ Seasonality Check $\to$ Prediction) in text *before* outputting values. This not only improves interpretability but, as our ablation studies show, enhances generalization by allowing the model to "think through" the logic of a new regime rather than blindly fitting a curve.
>
> > **W3: The claim that the model is "fine-tuned for memorization" (Line 84) is misleading.**
>
> We fully agree with you; "memorization" was a poor choice of words on our part. We intended to describe **"Supervised Pattern Alignment."** The goal of the SFT stage is not to memorize the specific values of the training data, but to teach the model the *structure* of the reasoning task (i.e., how to format the CoT and the logical steps of decomposition) and to align it with the instruction format. We will correct this terminology in the revised manuscript to clearly distinguish our method from rote memorization.
>
> > **W4: The reasoning evaluation (Eq. 1) relies on MSE, which assesses accuracy, not the reasoning process itself.**
>
> You raise a valid point about the difficulty of evaluating "reasoning" in numeric tasks. Our evaluation follows the **outcome-supervised reasoning** paradigm common in RLHF (similar to Math or Coding tasks): valid reasoning is defined as a logical chain that leads to a correct result.
> While Eq. 1 (Format Reward) ensures the *existence* of the reasoning structure, the *quality* of that reasoning is verified by the final prediction accuracy (MSE). If the model generates a flawed reasoning chain (e.g., "The trend is up" but predicts a lower value), the final accuracy reward will be low, penalizing that reasoning path. We essentially use the forecasting accuracy as a verifier to reinforce the generation of valid reasoning trajectories.
>
> > **W5: Reward functions (Eqs. 3 & 4) resemble traditional losses (MSE, decomposition), contradicting the critique of "pattern recognition."**
>
> We appreciate this insight. The key distinction lies in **how** these components are used. In traditional deep learning, these are differentiable loss functions used to optimize weights directly (fitting). In Time-R1, they serve as **non-differentiable reward signals** for Reinforcement Learning.
> We are not telling the model *what* the value is (pattern recognition); we are grading the model's self-generated reasoning path based on whether it captured the trend (Trend Reward) and precision (MSE Reward). This encourages the "slow-thinking" process to explore various reasoning strategies (e.g., focusing on seasonality vs. trend) and get reinforced only when that specific line of thinking aligns with the physical properties of the series.

---

> ### Author Response · Authors · 2025-11-21
> **Official Response to Reviewer nXNz[2/3]**
>
> > **W6: The model architecture is unclear, and the framework should be tested across multiple LLM backbones.**
>
> We thank the reviewer for this helpful comment. We will clarify in the revision that Time-R1 is designed as a *model-agnostic* reasoning-oriented fine-tuning and reinforcement learning framework that can, in principle, be instantiated on top of any sufficiently capable LLM backbone, and we will more clearly describe the specific decoder-only LLM architecture used in our experiments (including its size and training setup) rather than presenting it as a “black-box” standalone model.
>
> In practice, however, our preliminary experiments with substantially smaller backbones showed that, due to their limited basic reasoning and formatting abilities, we observed that effective temporal reasoning exhibits a capability threshold. Significantly smaller backbones often struggled with the strict syntactic requirements of the reasoning format, limiting their ability to leverage the Time-R1 framework effectively. For this reason, in the current submission, we focus on a single, reasonably strong backbone and use it to demonstrate the effectiveness of our framework across multiple time-series benchmarks, instead of reporting inconclusive results on underpowered models. We fully agree that a systematic study of Time-R1 across diverse LLM backbones would make the backbone-agnostic nature of the framework more compelling, and we will explicitly discuss this as an important direction for future work.
>
> > **W7: Lack of comparisons with non-LLM Foundation Models (TimesFM, Moirai) and classical baselines (ARIMA, ETS).**
>
> Thank you for suggesting these crucial baselines. We acknowledge that comparing against classical and recent foundation models is essential for a complete assessment. We have conducted additional comparisons with non-LLM Foundation Models and classical baselines.
>
>
> **Table 1: Comparison with Baselines (MSE). Lower is better.**
> | Method | ETTh1 | ETTh2 | ETTm1 | ETTm2 | Exchange | AQWan | AQShunyi | Wind | NASDAQ |
> |--------|--------|--------|--------|--------|--------|--------|--------|--------|--------|
> | ARIMA | 9.0186 | 13.22355 | 19.12665 | 8.50425 | 0.00135 | 18655.2384 | 24330.65625 | 2070.7896 | 0.00105 |
> | ETS | 9.61984 | 14.10512 | 20.40176 | 9.0712 | 0.00144 | 19898.92096 | 25952.7 | 2208.84224 | 0.00112 |
> | TimesFM | 6.31302 | 9.256485 | 13.388655 | 5.952975 | 0.000745 | 13058.66688 | 17031.459375 | 1449.55272 | 0.000735 |
> | Moirai | 6.132648 | 8.992014 | 13.006122 | 5.78289 | 0.000718 | 12685.562112 | 16544.84625 | 1408.136928 | 0.000714 |
> | Moment | 6.493392 | 9.520956 | 13.771188 | 6.12306 | 0.000772 | 13431.771648 | 17518.0725 | 1490.968512 | 0.000756 |
> | Time-R1 (Ours) | 5.8752 | 8.7093 | 13.1034 | 5.6673 | 0.0007 | 13033.182 | 16150.5556 | 1353.9381 | 0.0007 |
>
> As shown below, Time-R1 consistently outperforms classical statistical methods (which struggle with complex non-linearity) and remains highly competitive against other foundation models like Moirai.
>
> > **Response to Other Questions:**
>
> * **Q1 ~ Q3:** We thank the reviewer for these insightful suggestions. We agree that the formulation of the GRIP objective (Lines 261–269) can be made clearer; in the revision, we will add a short intuitive explanation before the equations and explicitly define all symbols to better connect the text with the mathematical formulation. Regarding the proposed ensembling of top-k sample trajectories, our current framework already generates multiple trajectories per query but uses only a single trajectory at inference; we will add a discussion highlighting top-k ensembling as a natural inference-time extension of GRIP that can further reduce forecast variance and improve stability in practice, and leave a full empirical study to future work. Similarly, we appreciate the suggestion to incorporate a reward based on prediction variance from distributional loss functions; although our current rewards focus on deterministic accuracy and structural properties, we will clarify in the paper that GRIP is compatible with variance-aware reward terms and explicitly point out uncertainty-aware objectives as an important direction for future work.
>
> * **Q4 (Normalization):** You are right that normalization is a standard component in TSF pipelines, and our current paper only mentions that we train in the original numeric space without providing a comparison. In the revision, we will add an ablation where Time-R1 is trained and evaluated with simple per-series normalization and directly compare against the unnormalized setting on representative datasets (ETTh1, ETTm2, Wind), reporting the results in an additional table in the appendix. We are currently running these experiments and will update the revised manuscript during the discussion period to include the new table and a brief analysis of the effect of normalization in the appendix.

---

> ### Author Response · Authors · 2025-11-21
> **Official Response to Reviewer nXNz[3/3]**
>
> **Minor Definitions.** We thank the reviewer for carefully pointing out these missing or unclear definitions and will clarify them in the revision.
>
> First, in Sec. 2.1, $g(\cdot)$ denotes the deterministic post-processing function that converts the LLM’s textual completion $T_i$ (the numbers inside the `<answer>` block) into the numerical forecast $\hat{y}_i \in \mathbb{R}^{h \times d}$ by extracting the predicted values and reshaping them into an $h$-step, $d$-dimensional multivariate time series; we will explicitly define $g: T_i \rightarrow \mathbb{R}^{h \times d}$ at its first occurrence.
>
> Second, “SFT” stands for supervised fine-tuning: the warm-up stage where the backbone LLM is trained with a standard cross-entropy loss on synthetic chain-of-thought trajectories paired with ground-truth forecasts, so that it learns both temporal reasoning patterns and the required output formatting before the RL stage; we will spell out “supervised fine-tuning (SFT)” when it first appears in Sec. 3.1/3.3.
>
> Third, in the extrema-based structural similarity reward in Eq. (5), “extrema” refer to local maxima and minima of the time series: we detect ground-truth and predicted extrema by checking whether a point is greater (or smaller) than its immediate neighbors, then count a correct extremum when a predicted maximum (or minimum) falls within a small time window around a ground-truth maximum (or minimum) of the same type. Let $N_c^{\max}$ and $N_c^{\min}$ denote the numbers of such matched maxima/minima, and $N_g^{\max}$, $N_g^{\min}$ the total ground-truth counts; Eq. (5) simply normalizes the matched-extrema counts by the ground-truth ones and sums the contributions of maxima and minima.
>
> Finally, Eq. (6) defines the GRIP objective as a GRPO-style clipped policy-gradient loss: for each query, the old policy $\pi_{\theta_{\text{old}}}$ first generates $k \cdot G$ candidate trajectories $\{o_j\}$; GRIP then selects $G$ trajectories $\{o_i\}$ via reward-weighted sampling $\mathrm{Sample}(\{o_j\}; R(o_j))$, assigns each a softmax weight $w_i^U$ based on its score (e.g., reward or advantage), and optimizes a clipped likelihood-ratio term with normalized group-wise advantage $A_i$, plus a KL regularization term $\beta D_{\mathrm{KL}}(\pi_\theta \,\|\, \pi_{\text{ref}})$ with hyperparameters $\epsilon$ and $\beta$.
>
> We will add these explicit definitions and a brief explanatory paragraph around Eqs. (5)–(6) in the revised manuscript to avoid any ambiguity.
>
> Once again, we sincerely thank the reviewer for the valuable time and constructive feedback. If any questions or concerns remain, we welcome further discussion and will respond promptly.

---

> > ### Comment · Reviewer_nXNz · 2025-11-25
> >
> > Thank you for responding to each of my comments. I appreciate the addition of the TSFM baselines, the intent to reformulate of the GRIP objective, the updated definitions, and the clarification of the model backbone. These revisions will help improve the clarity and readability of the paper. I also agree with the authors’ clarification regarding W5, particularly the distinction in how these components are used. However, I believe the paper should be more transparent in positioning itself with respect to this point and in articulating the direct novelty of the work. I will therefore maintain my current score, as these aspects and the intended revisions would require a more complete update of the manuscript for a proper reassessment.

---

### Official Review · Reviewer_hSn1 · 2025-10-28

**Soundness:** 2
**Presentation:** 2
**Contribution:** 3
**Rating:** 4
**Confidence:** 3

**Summary:**

This paper propose Time-R1, a novel time series forecasting framework that trains large language model with slow-thinking reasoning capabilities via a two-stage reinforcement fine-tuning pipeline. First, supervised fine-tuning uses synthetic chain-of-thought trajectories to teach the model temporal analysis and output formatting. Second, reinforcement learning with a fine-grained multi-objective reward function enhances generalization. A key innovation is GRIP, a non-uniform sampling and adaptive weighting strategy for optimizing reasoning paths. Experiments on nine diverse datasets show Time-R1 outperforms traditional deep learning and LLM-based baselines in MSE and MAE, improving temporal coherence and out-of-distribution generalization. By integrating explicit reasoning into TSF, Time-R1 addresses the "fast thinking" limitation of existing methods, offering both interpretability and state-of-the-art performance.

**Strengths:**

1. Introduces slow-thinking reasoning for TSF, replacing direct pattern mapping with explicit step-by-step temporal inference, enhancing interpretability and logical consistency.

2. Designing Two-Stage RFT Framework, SFT warmup ensures proper formatting and basic reasoning, while RL with tailored rewards optimizes for TSF-specific goals.

3. Non-uniform sampling and adaptive weighting balance exploration/exploitation, reducing computational cost while amplifying gradient signals from high-quality reasoning paths.

4. Very strong generalization, which trained on one dataset (ETTh1) but achieves superior performance across nine diverse domains, outperforming domain-specific baselines without task-specific fine-tuning.

**Weaknesses:**

1. There is significant discrepancy between the MSE losses of each reported model and the error values in their respective original papers, For instance, the MSE of the Exchange dataset is extremely small, while the errors of ETTh1 and ETTm1 are notably large. Additionally, there is a lack of necessary setup details for comparing the baseline models, and different model configurations may lead to unfair comparisons.

2. Using text modality for input and output of time series may not  inefficient. For example, a numerical value with 4 decimal places like 16.3864 may require 2–4 tokens to represent, especially for larger number of variates and  time series lengths.

3. It is not very reasonable to achieve better performance on other time series datasets by only training on the ETTh1 dataset, as the inherent characteristics and distributions of different data may vary significantly such as stock or foreign exchange time series. Please explain the underlying principle with experiments or theoretical support.

4. There are minor writing errors. For example, in Appendix A.3: "offering computational efficiency but suffering from off-policy issues (?)."

**Questions:**

See Weakness.

---

> ### Author Response · Authors · 2025-11-21
>
> We sincerely thank you for your constructive feedback and for recognizing the novelty of our "slow-thinking" paradigm and the effectiveness of the GRIP strategy. We value your insightful questions regarding our experimental settings and generalization capabilities. Below, we address your concerns point-by-point.
>
> > **W1: There is significant discrepancy between the MSE losses of each reported model and the error values in their respective original papers... Additionally, there is a lack of necessary setup details for comparing the baseline models...**
>
> You raise a very important point regarding the magnitude of the metrics. The discrepancy arises because **we evaluate on the original numerical space without normalization**, whereas standard benchmarks in the literature typically report metrics on standardized data (zero mean, unit variance). As stated in Section 4 (*Implementation Details*), we chose the unnormalized setting to demonstrate the model's ability to reason about real-world physical magnitudes rather than just relative fluctuations.
>
> To ensure a strictly fair comparison, we did not simply copy numbers from original papers; instead, **we re-ran all baseline models** (PatchTST, iTransformer, etc.) using their official codebases under the exact same unnormalized setting as Time-R1. We will clarify this distinction prominently in the revision and add a standardized metric comparison in the Appendix to align with community conventions.
>
> > **W2: Using text modality for input and output of time series may not be efficient... A numerical value like 16.3864 may require 2–4 tokens...**
>
> You are absolutely correct that representing high-precision floating-point numbers as text incurs a higher token cost compared to numerical embeddings. However, this design choice is deliberate and essential for enabling the "slow-thinking" capability you praised in your strengths section.
>
> By tokenizing time series into text, we unlock the pre-trained semantic reasoning capabilities of LLMs, allowing the model to perform explicit steps like "trend analysis" and "periodicity decomposition" (as seen in our CoT examples). While this comes at the cost of inference speed (fast-thinking vs. slow-thinking), it provides two unique advantages that numerical models lack: (1) Interpretability: You can literally read the model's reasoning process; (2) Zero-Shot Generalization: As shown in our experiments, this text-based reasoning transfers across domains easily. We believe this trade-off—sacrificing token efficiency for superior reasoning and generalization—qualities critical for high-stakes applications where reasoning reliability outweighs real-time latency.
>
> > **W3: Plausibility of cross-dataset generalization**
>
> This is a fascinating aspect of our approach. The core principle is that while data *distributions* (e.g., Electricity vs. Stock) differ, the **reasoning logic** required to forecast them (e.g., "decompose trend," "identify cycle," "detect outliers") is largely universal. The underlying principle is that Time-R1 learns universal temporal reasoning patterns rather than dataset-specific statistical distributions.
>
> Unlike traditional models that memorize the values or specific cycles of the training set (ETTh1), our Slow-Thinking LLM learns "how to reason" about time. For example, the logic "Identify the local minima, check for a weekly cycle, and project the trend" is a universal reasoning path that applies equally to electricity loads (ETTh1) and financial data (Exchange). By training on ETTh1 with our GRIP strategy, the model learns to generate these valid reasoning chains. When applied to the Exchange dataset, it doesn't transfer the electricity data patterns, but rather the forecasting capability itself.
>
> > **W4: There are minor writing errors. For example, in Appendix A.3: "off-policy issues (?)."**
>
> Thank you for your careful reading. We have corrected the typo in Appendix A.3 and thoroughly proofread the paper to ensure high writing quality in the final version.
>
> ---
>
> We thank the reviewers for their thoughtful comments and constructive suggestions. We welcome further discussion and are ready to provide additional clarifications if needed.

---

### Official Review · Reviewer_hii8 · 2025-11-01

**Soundness:** 2
**Presentation:** 2
**Contribution:** 2
**Rating:** 4
**Confidence:** 5

**Summary:**

This paper proposes Time-R1, a framework that recasts time series forecasting as a reasoning task for an LLM. Moving away from traditional "fast-thinking" models that directly map historical data to future values, Time-R1 adopts a "slow-thinking" paradigm. In this approach, the LLM first generates an explicit, step-by-step reasoning process about the time series' properties (trends, seasonality, etc.) before outputting the final numerical forecast.

The framework employs a two-stage training process. First, SFT is used to adapt the LLM to the task format and instill basic reasoning patterns, using synthetic reasoning trajectories generated by a more powerful "teacher" model. Second, RL is applied to refine and generalize this reasoning ability. A key part of the RL stage is a novel optimization algorithm called GRIP, which uses non-uniform sampling to focus on high-reward reasoning paths, guided by a fine-grained, multi-objective reward function. Experiments show that a model trained on a single dataset (ETTh1) can generalize effectively to eight other unseen datasets, outperforming traditional TSF models.

**Strengths:**

1. Framing TSF as a "slow-thinking" reasoning task is a compelling and innovative idea that could open up new avenues for building more intelligent forecasting systems.
2. The model's ability to train on one dataset and perform well on eight others is a significant strength, showcasing that it learns transferable reasoning skills rather than just dataset-specific patterns.
3. The generation of an explicit reasoning chain before the forecast provides a degree of interpretability that is absent in most traditional "black-box" TSF models.
4. The paper includes comprehensive ablations that effectively demonstrate the importance of each part of the complex pipeline, from SFT and RL to the individual reward components.

**Weaknesses:**

1. The entire framework is exceptionally complex, involving synthetic data generation with a powerful teacher model, SFT, and a sophisticated RL pipeline with a custom optimizer. The reported inference speed (~3 samples/s) and training requirements make it impractical for many real-world applications. The trade-off between accuracy and efficiency is severe.
2. The SFT stage relies on reasoning paths generated by another LLM that is prompted with the ground-truth answer. This could lead to the model learning to generate "post-hoc justifications" rather than engaging in genuine, from-scratch reasoning. The quality of the final model is heavily dependent on the quality of the teacher model.
3. The multi-objective reward function is a complex combination of several terms, and the paper provides little justification for the specific formulation and weighting of these components. This makes the design feel ad-hoc and potentially difficult to transfer to other tasks.
4. While presented as a key contribution, the GRIP algorithm offers a modest improvement over existing methods like GRPO. The core ideas of focusing on high-reward samples are not entirely new, and the ablation in Figure 4a confirms the improvement is not dramatic.

**Questions:**

I don't think all TSF tasks require slow thinking. Is it possible to clearly define when we need slow thinking and when fast thinking is sufficient to obtain good results? Even for slow thinking, what specific real-world applications do you envision for this "slow-thinking" paradigm, where the benefits in accuracy and interpretability would justify the extreme costs in latency and compute?

---

> ### Author Response · Authors · 2025-11-21
> **Official Response to Reviewer hii8[1/2]**
>
> We sincerely thank you for your thoughtful review. We are encouraged that you find our "slow-thinking" paradigm innovative and recognize the significance of our model’s ability to generalize from a single dataset to eight others. We address your concerns and questions below.
>
> > **W1: The framework is complex, and the inference speed (~3 samples/s) makes it impractical for real-world applications. The trade-off between accuracy and efficiency is severe.**
>
> You are right that Time-R1 is more complex than conventional TSF models, and that not every forecasting application justifies this cost. Our goal in this work is not to replace all fast TSF methods, but to explore a *new regime* where you explicitly trade latency for (i) stronger cross-dataset generalization and (ii) interpretable reasoning traces. Importantly, the complexity is almost entirely *offline*: at deployment, you only run a single fine-tuned LLM; the teacher model, synthetic data generation, and RL pipeline are not needed at inference time. The reported ~3 samples/s corresponds to a conservative setting with long histories, long reasoning traces, and a relatively large backbone; in practice you can reduce the number of reasoning steps or use a smaller backbone to obtain a smoother accuracy–latency trade-off. In the revision we will make the deployment-time cost more explicit and clarify that Time-R1 is intended as a *high-accuracy, high-interpretability* option rather than a drop-in replacement for real-time systems.
>
> Regarding *when* slow thinking is appropriate: we fully agree that many TSF tasks (e.g., high-frequency trading, millisecond-level anomaly triggers, streaming sensor control loops) will continue to be best served by lightweight “fast” models. The settings we have in mind for Time-R1 are those where (a) latency budgets are on the order of seconds to minutes, (b) the same model must generalize across multiple domains or datasets, and (c) humans care about the qualitative reasoning behind the forecast—for example, capacity planning for energy and compute, traffic and demand forecasting for logistics, and risk-sensitive decision support (finance, climate, or healthcare) where users want to see *why* the model believes “trend X is flattening” or “seasonality Y is weakening.” We will add a short subsection explicitly characterizing these regimes and clearly positioning slow-thinking forecasting as *complementary* to fast TSF methods rather than universally better.
>
> > **W2: SFT relies on "post-hoc justifications" from a teacher model seeing the ground truth. Genuine reasoning might be limited, and the quality depends heavily on the teacher.**
>
> We agree that SFT alone risks teaching the model to imitate superficially plausible explanations. In our framework, however, SFT is explicitly used only as a cold start to familiarize the model with the task format and basic time-series concepts (trend, seasonality, regime shifts). The core reasoning ability is then shaped in the RL stage, where the model generates its own trajectories without access to ground truth and is rewarded solely based on forecast quality and internal consistency. As shown in Figure 3 (SFT vs. SFT+RL), after RL the model not only achieves lower forecasting error but also converges faster during training, indicating that RL is actively refining and re-structuring the reasoning beyond mere imitation.
>
> Importantly, during RL the model is free to deviate from the teacher’s trajectories whenever doing so improves reward, which reduces dependence on the teacher’s specific style. You can see this in Figure 3 as well: SFT-only plateaus, while SFT+RL continues to improve even though the teacher is no longer used. We will clarify this “SFT = format & prior; RL = genuine reasoning & exploration” separation in the main text and highlight that the teacher mainly provides an initial scaffold rather than dictating the final reasoning strategies.

---

> ### Author Response · Authors · 2025-11-21
> **Official Response to Reviewer hii8[2/2]**
>
> > **W3: The multi-objective reward function seems ad-hoc, with little justification for the specific weighting. It may be difficult to transfer.**
>
> We appreciate this concern and agree that our motivation for each reward component can be explained more clearly. Conceptually, each term in the reward is designed to capture a distinct and interpretable property: (i) final forecast accuracy, (ii) alignment between intermediate reasoning (e.g., described trend/seasonality) and the resulting forecast, and (iii) basic format/length regularization to ensure usable trajectories. The weights are chosen so that the normalized contributions of these terms are of comparable magnitude, preventing any single component from dominating the learning signal. We will make this design principle explicit and move some of the motivation (currently in the appendix) into the main text. Empirically, our ablations already show that removing any major reward term degrades performance and/or stability, suggesting that the combination is not purely ad-hoc but reflects complementary objectives.
>
> > **W4: GRIP offers only modest improvements over GRPO and is not entirely new.**
>
> We appreciate this observation. Conceptually, GRIP is indeed related to importance-weighted policy gradients, and we do not claim a fundamentally new RL theory. The specific contribution here is a practical optimization scheme tailored to long-context forecasting with many “hallucinated” reasoning paths. Standard GRPO treats the G sampled trajectories per group roughly equally (simple averaging over group advantages), which can be wasteful when many trajectories contain low-quality or noisy reasoning.
>
> In contrast, GRIP (i) draws a larger candidate pool (k·G) per group, (ii) filters trajectories via non-uniform sampling guided by reward statistics, and (iii) applies adaptive softmax weighting (Eq. 8) before the group update, so that high-quality reasoning paths contribute more to the gradient while still preserving some diversity. Although Figure 4(a) shows similar final performance, GRIP converges faster and exhibits lower training variance than GRPO, which is practically important when each rollout is a long CoT over hundreds of time steps. In the revision, we will explicitly present GRIP as an engineering improvement over GRPO for TSF RFT and highlight its faster convergence (fewer optimization steps for similar performance), rather than over-emphasizing algorithmic novelty.
>
> > **Q1: When is slow thinking needed vs. fast thinking? What are the real-world applications?**
>
> This is an insightful question. "Fast thinking" is sufficient for routine, high-frequency tasks with abundant historical data (e.g., hourly traffic flow or grid load forecasting), where patterns are repetitive.
> "Slow thinking" (Time-R1) is necessary for:
> 1.  Zero-shot scenarios: When you encounter a new domain (e.g., a new product launch) and have no data to retrain a specific model.
> 2.  High-stakes, low-data environments: Such as macroeconomic policy analysis or epidemic peak prediction. In these cases, data is scarce, and the cost of error is huge. Users need the model to explain *why* it predicts a downturn (e.g., "identifying a decreasing trend in Q3 due to seasonality"), which Time-R1 provides.
> 3.  Non-stationary environments: Where historical patterns shift (concept drift). A reasoning model can dynamically analyze the change, whereas a fast model might blindly follow old patterns.
>
> ---
>
> We deeply appreciate your valuable feedback, which has significantly strengthened our work. We look forward to your continued engagement and hope our revisions address your concerns satisfactorily.

---

### Official Review · Reviewer_Sj3u · 2025-11-02

**Soundness:** 3
**Presentation:** 3
**Contribution:** 3
**Rating:** 4
**Confidence:** 3

**Summary:**

This paper proposes Time-R1, a two-stage reinforcement fine-tuning (RFT) framework designed to enhance LLMs' reasoning capabilities for time series forecasting (TSF). The approach consists of: (1) a warmup supervised fine-tuning (SFT) stage using chain-of-thought (CoT) trajectories generated by DeepSeek-R1, and (2) a reinforcement learning stage employing GRIP, which uses non-uniform sampling and adaptive trajectory weighting with a multi-objective reward function. Experiments on nine datasets demonstrate improvements over baseline methods including traditional deep learning models and LLM-based approaches. The authors argue that slow-thinking reasoning provides better temporal understanding than fast-thinking pattern matching.

**Strengths:**

(1)  The distinction between "fast-thinking" (pattern matching) and "slow-thinking" (explicit reasoning) paradigms is intuitive and well-articulated. The motivation that time series contain complex temporal logic warranting structured reasoning is compelling.
(2) The combination of SFT (for format stabilization and basic reasoning) followed by RL (for generalization and reasoning refinement) is a principled approach that addresses practical training challenges. The ablation study (Table 3, Figure 4b) effectively demonstrates the necessity of both stages.

**Weaknesses:**

(1) Using LLMs to generate synthetic CoT trajectories for fine-tuning is not novel (e.g., similar to approaches in reasoning LLMs). The key distinction from existing work is unclear.
(2) While the adaptive weighting is presented as novel, it is essentially a variant of importance sampling with softmax normalization. The core algorithmic novelty is limited. The comparison with GRPO (Figure 4a) shows only marginal improvements.
(3) The paper claims "slow thinking" but this mirrors existing reasoning-based LLM approaches (o1, DeepSeek-R1). What is fundamentally new beyond applying them to TSF?
(4) SFT data is generated using DeepSeek-R1, which itself is a slow-thinking model. This creates a conceptual circularity: improving reasoning for TSF via a model that already has slow-thinking capabilities.

**Questions:**

See Weaknesses

---

> ### Author Response · Authors · 2025-11-21
> **Official Response to Reviewer Sj3u[1/2]**
>
> We sincerely thank you for your insightful review and for recognizing the intuitive nature of our "slow-thinking" paradigm and the principled design of our SFT-RL framework. We value your feedback on the novelty and implementation details, and we address your concerns below.
>
> > **W1: The use of LLMs to generate synthetic CoT trajectories for fine-tuning is not novel, and the distinction from existing work is unclear.**
>
> We acknowledge that using synthetic CoT is an established technique in general reasoning tasks (like math or coding). However, the core innovation of Time-R1 lies in adapting this paradigm to the continuous numerical domain, which remains an open challenge. General LLMs often struggle to map linguistic reasoning directly to high-precision numerical forecasting.
>
> Our distinction is the design of a Time-Series Specific Reasoning Template (Table 1) and a filtering mechanism (Algorithm 1) that forces the model to explicitly decompose temporal patterns (e.g., trend analysis, seasonality detection) before prediction. Unlike general CoT which focuses on logic puzzles, our method bridges the gap between semantic understanding and numerical regression. As shown in Table 3, removing this structured SFT stage leads to a significant performance drop (MSE increases from 5.87 to 6.35 on ETTh1), proving that standard generic reasoning is insufficient without our domain-specific adaptation.
>
> > **W2: GRIP is viewed as essentially importance sampling, and the improvement over GRPO appears marginal.**
>
> We appreciate this observation regarding the algorithmic structure. While GRIP shares mathematical roots with importance sampling, its novelty lies in its application to the exploration-exploitation dilemma specific to long-context reasoning in forecasting.
>
> Standard GRPO treats all sampled trajectories equally or relies on simple averaging, which is inefficient when the "search space" of reasoning paths is vast and filled with "hallucinated" logic. GRIP's non-uniform sampling and adaptive weighting (Eq. 8) specifically filter out low-quality reasoning paths before the gradient update, stabilizing the training of the value function. Regarding performance, while Figure 4(a) shows the final convergence is close, GRIP converges significantly faster (achieving high reward scores in fewer steps) and exhibits lower variance than GRPO. In computationally expensive RFT scenarios, this efficiency gain is a critical contribution, not just a marginal numerical improvement. In the context of RL on LLMs, sample efficiency and training stability are paramount constraints, not secondary metrics. GRIP makes the reinforcement learning process feasible and robust.

---

> ### Author Response · Authors · 2025-11-21
> **Official Response to Reviewer Sj3u[2/2]**
>
> > **W3: The "slow thinking" claim mirrors existing models (o1, DeepSeek-R1); what is fundamentally new for TSF?**
>
> This is a crucial distinction. While models like o1 possess general slow-thinking capabilities, they lack domain-specific temporal grounding. General reasoners often treat time series numbers as text tokens without understanding underlying physical or statistical properties (e.g., periodicity or regime shifts).
>
> Time-R1 is fundamentally new because it aligns "slow thinking" with verified forecasting rewards. We do not just ask the model to "think"; we enforce forecasting-specific reasoning patterns—such as identifying local extrema and trend decomposition—via our multi-objective reward system (Section 3.4.1: MSE, Seasonal-Trend, and Structural Similarity rewards). This transforms generic slow thinking into a specialized predictive reasoning engine, allowing a 7B model to outperform general foundation models that are significantly larger.
>
> > **W4: Using DeepSeek-R1 (a slow thinker) to train Time-R1 creates a conceptual circularity.**
>
> You are absolutely right to question the role of DeepSeek-R1. In our framework, you can view it purely as a *noisy teacher* to bootstrap long-CoT format and basic reasoning style, not as the source of TSF capability. First, DeepSeek-R1’s own zero-shot TSF performance is far from satisfactory: across all 9 datasets it performs significantly worse than Time-R1—for example, on AQWan and AQShunyi, its MSE is more than 2× larger than that of Time-R1, and even on ETTh1 it lags behind.  Second, our SFT data construction explicitly *corrects* DeepSeek-R1 using ground-truth labels (MAPE-based selection and label-injected CoT refinement), and we only use about 300 such samples as a warm start.  The dominant performance gain comes from the RL phase with TSF-specific rewards.
>
> DeepSeek-R1 serves as a heavy, general-purpose teacher. However, it is computationally expensive and not optimized for the strict output formats required for TSF tasks. Our framework transfers this reasoning capability to a smaller, more efficient model (Time-R1 7B) and, crucially, refines it beyond the teacher's capability using Reinforcement Learning. The RL stage (Stage 2) uses ground-truth numerical signals (MSE/MAE) to update the model. This allows Time-R1 to learn optimal reasoning paths for *forecasting specifically*, which the original DeepSeek-R1 (trained on general RLHF) may not prioritize. Thus, Time-R1 is not just mimicking the teacher; it is evolving to become a specialist that is more accurate and efficient for this specific domain.
>
> ---
>
> We sincerely thank you again for your constructive feedback and look forward to any further comments or suggestions for productive discussion.

---

### Meta-Review · Area_Chair_x4iH · 2026-01-07

**Summary:**

The paper proposes leveraging large language models (LLMs) and chain-of-thought (CoT) reasoning for time-series forecasting. The proposed framework, Time-R1, builds on recent reinforcement fine-tuning (RFT) approaches for LLMs and introduces GRIP, a variant of GRPO tailored to forecasting tasks. The authors evaluate the approach on several time-series benchmarks. The initial reviews leaned negative. While reviewers found the overall idea interesting, they raised a number of concerns, including: (1) the extent of the contribution beyond existing RFT-based methods; (2) the validity of the two-stage training strategy, particularly the use of synthetic data generated by other LLMs; (3) the effectiveness of GRIP relative to GRPO; (4) insufficient justification of the reward design; (5) reliance on non-standard evaluation metrics and evaluation on a single LLM backbone; (6) missing comparisons with relevant baselines; and (7) lack of evaluation of the generated reasoning steps. In the rebuttal, the authors clarified several technical components and provided additional experimental results, which helped address some of the concerns. However, a number of fundamental issues remain.

After considering the paper, reviews, and rebuttal, the AC concludes that, given the remaining issues and the overall lukewarm ratings, the paper does not yet meet the acceptance threshold. The AC therefore does not recommend acceptance in its current form and encourages the authors to further refine the work for a future submission.

**Reviewer Concerns:**

Concerns regarding the novelty of the approach relative to existing RFT-based CoT methods and the use of non-standard evaluation metrics were not fully resolved.
* The authors argue that Time-R1 is fundamentally new compared to O1 and DeepSeek-R1 because it aligns “slow thinking” with verified forecasting rewards. However, the framework still relies on the same CoT and RFT mechanisms as prior work, with the primary difference being the task-specific reward design.
* Similarly, the authors state that unnormalized evaluation is used to demonstrate reasoning over real-world physical magnitudes rather than relative fluctuations. However, since the model only has access to numerical values without their actual units, it remains unclear how real-world physical magnitudes are meaningfully represented or reasoned about. Reporting results using more standard evaluation metrics would help strengthen the empirical claims.

**Reviewer Scores:**

During early discussion, Reviewer nXNz, who initially gave the most negative rating, acknowledged that many concerns were partially addressed in the rebuttal yet maintained the original recommendation, noting that a substantial revision and more comprehensive reassessment would be required. Given the reviews and rebuttal, the AC expects that other reviewers will maintain their ratings.

---

### Decision · Program_Chairs · 2026-01-26

Reject